# SHORT-TERM MEMORY IN NEURAL LANGUAGE MODELS

## ABSTRACT

When a language model is trained to predict natural language sequences, its prediction at each moment depends on a representation of prior context. Thus, language models require mechanisms to maintain and access memory. Although we design the architectural features of these models, we do not know how their memory systems are functionally organized via learning: what kind of information about the prior context can they retrieve? We reasoned that access to arbitrary individual tokens from the past could be computationally powerful, akin to the working memory which is important for flexible cognition in humans, and we therefore tested whether language models could "retrieve" the exact words that occurred previously in a text. In particular, we tested how the ability to retrieve prior words depended on (i) the number of words being retrieved, (ii) their semantic coherence, and (iii) the length and quality of the intervening text. We evaluated two particular architectures of neural language models: the attention-based transformer and the long short-term memory network (LSTM). In our paradigm, language models processed English text in which a list of nouns occurred twice. We operationalized retrieval as the reduction in surprisal from the first presentation of the list to its second presentation. We found that the transformer models retrieved both the identity and ordering of nouns from the first list. The transformer was successful even when the noun lists were semantically incoherent, and this effect was largely robust to the type or length of the intervening text. Further, the transformer's retrieval was markedly enhanced when it was trained on a larger corpus and with greater model depth. Lastly, its ability to index prior tokens was dependent on learned attention patterns. In contrast, the LSTM models exhibited less precise retrieval (smaller reductions in surprisal). The LSTM's retrieval was limited to list-initial tokens, and occurred only across short intervening texts. Moreover, the LSTM's retrieval was not sensitive to the order of nouns and this non-specific retrieval improved when the list was semantically coherent. In sum, the transformer, when trained to predict linguistic tokens, implements something akin to a working memory system, as it could flexibly retrieve individual token representations across arbitrary delays. Conversely, the LSTM maintained a coarser and more rapidly-decaying semantic gist of prior tokens, weighted heavily toward the earliest items. Thus, although the transformer and LSTM architectures were both trained to predict language sequences, only the transformer learned to flexibly index prior tokens.

## 1 INTRODUCTION

Language models (LMs) are computational systems trained to predict upcoming tokens based on past context. To perform this task well, they must construct a coherent representation of the text, which requires establishing relationships between words that occur at non-adjacent time points. Despite their simple learning objective, LMs based on contemporary artificial neural network architectures perform well on language processing tasks that require maintenance and retrieval of context spanning multiple words in a sentence. For example, LMs learn to correctly match the grammatical number of the subject and a corresponding verb across multiple intervening words; for example, they prefer the correct *The **girls** standing at the desk **are** tall*, to the incorrect *The **girls** standing at the desk **is** tall* (Linzen et al., 2016; Marvin & Linzen, 2018; Gulordava et al., 2018; Futrell et al., 2018). Moreover, language representations learned by large-scale LMs can be fine-tuned or directly used

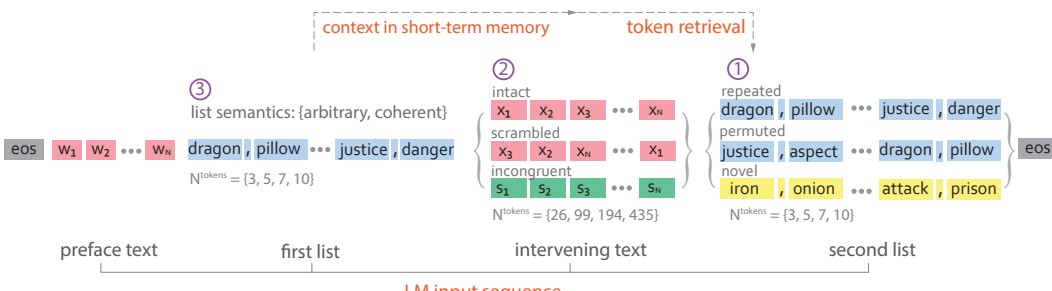

Figure 1: Assessing memory retrieval in neural language models. In language, memory of past context is required for ongoing comprehension. How are memory systems of neural LMs organized, via learning, for processing? In our paradigm, language models processed English text in which a list of nouns occurred twice. We operationalized retrieval as the reduction in surprisal from the first presentation of the list to its second presentation. The two lists of nouns were embedded in a vignette which consisted of a context string, first list, an intervening text, and the second noun list. To probe the properties of neural LM memory, we measured LM surprisal on the second list while varying: a) set size, b) the structure of the second list, c) the length of the intervening text, and d) the content and structure of the intervening text.

for solving downstream natural language processing (NLP) tasks (Devlin et al., 2019; Brown et al., 2020). Together, these results suggest that LMs extract linguistically meaningful signals and that, over the course of learning, they develop a short-term memory capacity: the ability to store and access recent past context for processing. But what is the nature of the memory processes that LMs learn? Are these memory processes able to access individual tokens from the recent past *verbatim*, or is the memory system more implicit, so that only an aggregate *gist* of the prior context is available to subsequent processing? The ability to access arbitrary individual tokens from the past could be computationally powerful, akin to the "working memory" systems thought to enable flexible human cognitive capacities (Baddeley, 2003).

In this work, we introduce a paradigm for characterizing the short-term memory abilities of a language model. We apply it to two particular architectures of neural language models that possess the architectural ingredients to hold past items in memory: attention-based transformers (Vaswani et al., 2017) and long short-term memory networks (Hochreiter & Schmidhuber, 1997, LSTM). Whereas an LSTM incorporates the past by reusing the results of processing from previous time steps and through dedicated memory cells, the transformer receives the internal representations of each of the previous tokens as input. These architectural ingredients alone, however, are not sufficient for a model to have memory. Whether or not the model puts this memory capacity to *use* depends on whether the training task (next word prediction) requires it — the parameters controlling the activation of context representations and subsequent retrieval computations are in both cases *learned*.

Our goal was to determine whether and when the LMs we study maintain and retrieve verbatim representations of individual prior tokens. First, we measured the *detail* of the context representation: does the LM maintain a verbatim representation of all prior tokens and their order, or does it instead combine multiple prior tokens into a summary representation, like a semantic gist? Second, we considered the *resilience* of the memory to interference: after how many intervening tokens do the representation of prior context become inaccessible? Third, we considered the *content-invariance* of the context representations: does the resilience of prior context depend on semantic coherence of the prior information, or can arbitrary and unrelated information sequences be retrieved?

Our paradigm, which is inspired by benchmark tasks for models of human short-term memory (Oberauer et al., 2018), reveals whether or not and under what conditions the LM can encode and

retrieve lists of nouns embedded in vignettes (see Fig. 1). By comparing LM surprisal on repeated lists of nouns, we found that:

- **The transformers we tested can perform verbatim recall of past nouns and are robust to interference.** The transformers retrieved precise representation of nouns (word identity and word order) from the context and did so over long-range intervening text (> 400 tokens).

- **The LSTM we tested stores a semantic gist of past nouns over short distances.** The LSTM's noun list retrieval capacity was limited to a semantic summary of the earlier nouns in the list, did not preserve the order or identity of the original nouns, and only succeeded across a small number of intervening tokens.

- **Verbatim recall in transformers improves with training corpus size.** The transformer trained on a larger dataset showed much larger effects compared to transformer trained on smaller corpus.

- **Verbatim recall in transformers requires depth.** Memory retrieval capacity emerged with 6- and 12- layer models, but not in smaller 1- and 3-layer models.

- **Order-sensitivity of transformers relies on learned attention patterns.** After randomly permuting attention weights, the transformer no longer showed ordering retrieval capacity.

## 2 RELATED WORK

Previous studies examined how properties of linguistic context influenced average LM performance. Khandelwal et al. (2018) showed that LSTM LMs use a window of approximately 200 tokens of past context and word order information of the past 50 words, in the service of predicting the next token in natural language sequences. Subramanian et al. (2020) applied a similar analysis to a transformer LM and showed that LM loss on test-set sequences was not sensitive to context perturbations beyond 50 tokens. These studies focused on how specific features of past context affect LM performance on novel input in the test set. In contrast, our paradigm tests for the ability of LMs to exactly retrieve nouns that are repeated from prior context.

## 3 METHODS

### 3.1 PARADIGM: LISTS OF NOUNS IN CONTEXT

Noun lists were embedded in brief vignettes (Figure 1, A and B). Each vignette opened with a *preface string* (e.g. "Before the meeting, Mary wrote down the following list of words:"). This string was followed by a list of nouns (the *first list*), which were separated by commas; the list-final noun was followed by a full stop (e.g. "county, muscle, vapor."). The first list was followed by an *intervening text*, which continued the narrative established by the preface string ("After the meeting, she took a break and had a cup of coffee."). The intervening text was followed by a short *prompt* string (e.g. "After she got back, she read the list again:") after which another list of nouns, either identical to the first list or different from it, was presented (we refer to this list as the *second list*).

### 3.2 SEMANTIC COHERENCE OF NOUN LISTS

We used two types of word lists: arbitrary and semantically coherent (all word lists are shown in Tables 1 and 2 in the Appendix). Arbitrary word lists (e.g. "device, singer, picture") were composed of randomly sampled nouns from the Toronto word pool.[1] Semantically coherent word lists were sampled from the categorized noun word pool,[2] which contains 32 lists, each of which contains 32 semantically related nouns (e.g. "robin, sparrow, heron, ...").

We ensured that the nouns in each list were part of the LSTM's vocabulary. After ensuring there were at least 10 valid, in-vocabulary nouns per semantic set (as this was the maximal list length we considered), we were able to construct 23 nouns lists. Finally, to reduce the variance attributable to tokens occurring in specific positions, we generated 10 "folds" of each list by circularly shifting the

---

[1] http://memory.psych.upenn.edu/files/wordpools/nouns.txt
[2] http://memory.psych.upenn.edu/files/wordpools/catwpool.txt

tokens in the first list 10 times. In this way, each noun in each list was tested in all possible ordinal positions. This procedure resulted in a total of $23 \times 10 = 230$ noun lists.

## 3.3 LANGUAGE MODELS

**LSTM** We used an LSTM released by van Schijndel et al. (2019),[3] which had 2 hidden layers with 400-dimensional hidden states each and had a vocabulary of 28,439 entries. The model contained 25.3 million trainable parameters. It was trained on a 40-million word subset of the Wikitext-103 corpus (Merity et al., 2016) and had a test-set perplexity of 79.6. The model's vocabulary contained 28,439 entries. We selected this checkpoint as our starting point because van Schijndel et al. (2019) showed that increasing the number of model parameters and size of training set beyond this point did not significantly improve the model's performance on the subject-verb agreement task. To test for the potential role of model/training set size on our task, we also evaluated a larger, 2-layer LSTM with 1600-dimensional hidden states (132 million trainable parameters), trained by van Schijndel et al. (2019) on 80 million tokens which had a test-set perplexity of 108.5.

**Transformer** We trained a transformer LM on $\sim 40$ million subword tokens of the Wikitext-103 benchmark (i.e. comparable to the corpus size used for training the LSTM checkpoint). We ensured that the vocabulary size was matched to the size of the LSTM vocabulary (28,439 entries) by retraining the tokenizer (see Appendix section A.4.2). We trained both the 12-layer GPT-2 architecture (known as "GPT-2 small", 107.7 million trainable parameters) and, as a point of comparison, smaller, 1-, 3-, and 6-layer transformers (29.7, 43.9, and 65.2 million trainable parameters, respectively). The context window was set to 1024 tokens and embedding dimension was kept at 768 across the architectures (see Section A.4 for training details). The perplexities for the 12-, 6-, 3- and 1-layer models on the Wikitext-103 test set were 40.3, 46.7, 60.1, and 93.2, respectively. Unless indicated otherwise, the label "Wikitext-103 transformer" refers to the 12-layer checkpoint in the figures.

We also evaluated the pretrained transformer LM released by Radford et al. (2019), which we refer to simply as GPT-2. This model was trained on the WebText corpus, which consists of approximately 8 million online documents. We used the GPT-2-small checkpoint which has 12 attention layers and 768-dimensional embedding layer. The model contains 124 million parameters and has a vocabulary of 50,257 entries. We used the maximum context size of 1024 tokens.

## 3.4 SURPRISAL

For each token $w_t$ in our sequence, we computed the negative log likelihood (surprisal): $\texttt{surprisal}(w_t) = -\log_2 P(w_t|w_1, \ldots, w_{t-1})$. In cases when the transformer byte-pair encoding tokenizer split a noun into multiple tokens—e.g. "sparrow" might be split into "sp" and "arrow"—we summed the surprisals of the resulting tokens.

**Quantifying retrieval: repeat surprisal** To quantify how the memory trace of the first list affected surprisal on the second list, we measured the ratio between the surprisal on the second list and the surprisal on the first list: $\texttt{repeat surprisal} = \frac{\bar{s}(L_2)}{\bar{s}(L_1)} \times 100$ where $\bar{s}(L_1)$ refers to mean surprisal across non-initial nouns in the first list and $\bar{s}(L_2)$ to median surprisal across all non-initial nouns in the second list. We take a *reduction* in surprisal on second lists to indicate the extent to which an LM has retrieved tokens from the first list.

## 4 TRANSFORMER RESULTS

### 4.1 THE TRANSFORMERS RETRIEVED PRIOR NOUNS AND THEIR ORDER; THIS CAPACITY IMPROVED WHEN THE MODEL WAS TRAINED ON A LARGER CORPUS

We tested whether the transformers could retrieve the identity and order of 10-token noun lists (arbitrary or semantically coherent). To this end, we constructed vignettes in which the second list

---

[3]Models are available at: `https://doi.org/10.5281/zenodo.3559340`

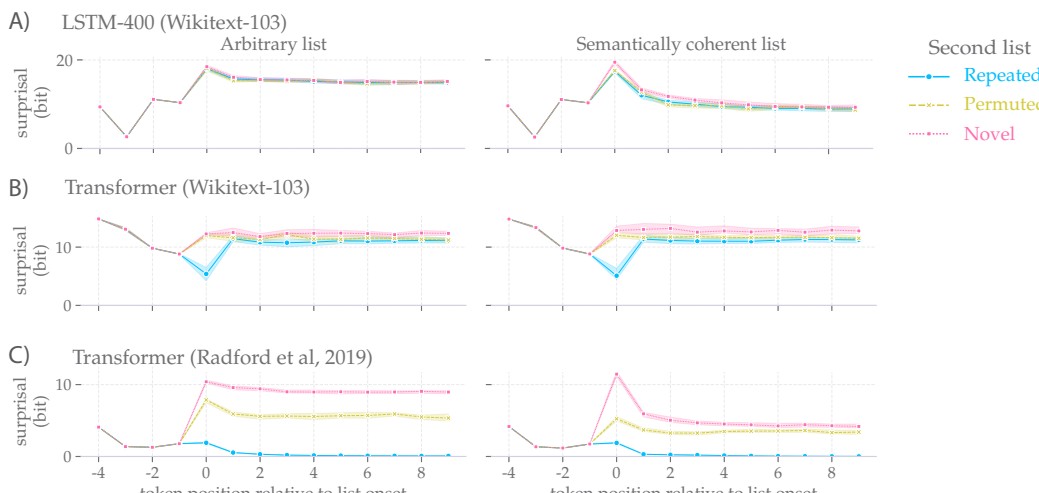

Figure 2: Median surprisal (over $N^{list} = 230$) broken down per token position in second lists of arbitrary nouns and semantically coherent nouns. Negative values on x-axis represent 4 tokens of prompt string that introduced the second list: "(she) read the list again". The 0-index marks the first noun in the list. Line style and hue denote manipulation of the second list relative to the first list. Error bands denote 95% confidence interval around the median (bootstrap estimate).

was either a) identical to the first list, b) a permutation of the first list, or c) a list of novel nouns not present in the first list.[4] We then measured retrieval as reduction in surprisal from first to second list.

When the two transformers were presented with second lists that were repeated version of the first, token-by-token (blue in Fig. 2, B and C) surprisal decreased compared to novel tokens, suggesting that the transformer was able to access verbatim representation of past nouns from context. When the second list was a permutation of the first one, surprisal was higher compared to when it was repeated, indicating that the transformer expected the nouns to be ordered as in the first list. Importantly, surprisal differences were considerably smaller for the transformer trained on the 40 million Wikitext-103 corpus (Fig. 2, B) compared to GPT-2 (Fig. 2, C). This suggests that the training set size plays an important role in supporting verbatim recall.

In order to contextualize the magnitude of these retrieval effects, we computed the relative surprisal (Section 3.4) across all tokens in lists except the first one (Fig. 3, left). When the first and second lists were identical, the Wikitext-103 transformer relative surprisal was $\sim 80\%$ of the first list, lower than the $\sim 85\%$ for the permuted lists, and $\sim 100\%$ for the novel list. In GPT-2, repeat surprisal was $\sim 2\%$ of the first list, much lower than the $\sim 60\%$ for the permuted lists, and $\sim 100\%$ of the novel list.

Retrieval in GPT-2 was robust to the exact phrasing of the text that introduced the lists. Replacing the subject 'Mary' with 'John' in the vignette, replacing the colon with a comma or randomly permuting the preface or the prompt strings did not affect the results (see Fig. 9 in A.8). The same control analyses show that the retrieval effects for Wikitext-103 transformer were reduced under these variations (see Fig. 8 in A.8) confirming that larger corpus size contributes to robustness of the transformer retrieval.

### 4.2    TRANSFORMER RETRIEVAL WAS ROBUST TO THE NUMBER OF ITEMS BEING RETRIEVED

In studies of human short-term memory, performance degrades as the number of items that need to be retained increases ("set-size effects", Oberauer et al. 2018). Is our LMs' short-term memory

---

[4]Novel nouns in the string were introduced by randomly selecting a list of nouns from one the 22 remaining lists in the noun pool. In semantically coherent lists, novel nouns were drawn from a different semantic category than the nouns in the first list.

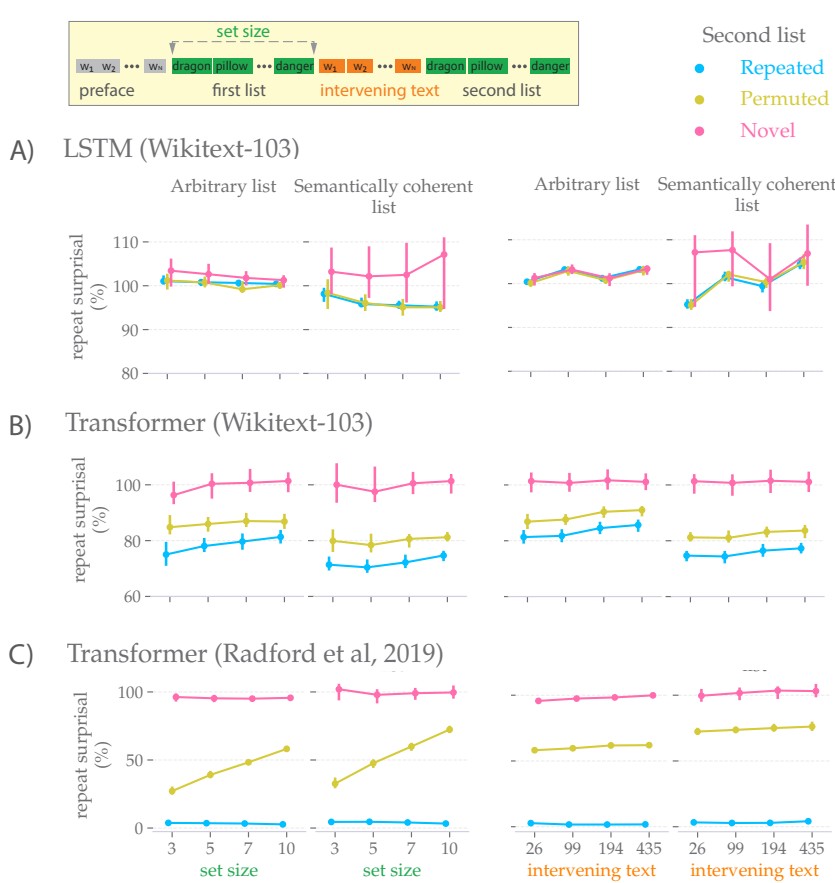

Figure 3: Verbatim token retrieval for varying number of tokens being retrieved (left) and the length of the intervening text (right). Reported is proportion of list-median surprisal on second relative to first list of nouns. Points show group median (over $N^{list} = 230$). Error bars denote 95% confidence interval around the median (bootstrap estimate). For set size manipulation, intervening text is fixed at 26 tokens. For intervening text manipulation, set size is fixed at 10 tokens.

similarly taxed by increasing the set size? We varied the number of tokens to be held in memory with $N^{tokens} \in \{3, 5, 7, 10\}$. For this comparison, the length of intervening text was kept at 26 tokens. Results reported in Fig. 3 show that for both the smaller Wikitext-103 transformer and the larger GPT-2, verbatim recall was, for the most part, consistent across the different set sizes. For GPT-2, repeat surprisal increased monotonically with set size only when the order of nouns in second list, either semantically coherent or arbitrary, was permuted. This increase in surprisal with set size for permuted sequences occurred because, if the model has perfect memory of the list of tokens, but cannot predict the order in which they will reoccur, then its probability of guessing the next item in a permuted list of $n$ possible tokens will be $1/n$, and the mean value of $n$ is larger for larger set sizes.

## 4.3 TRANSFORMER WORD RETRIEVAL WAS ROBUST TO THE SIZE AND CONTENT OF INTERVENING TEXT, BUT SCRAMBLING THE INTERVENING TEXT REDUCED RETRIEVAL OF ORDER INFORMATION

For how long are individual items retained in the memory of the LM? We tested this by varying the length of the intervening text for $N^{tokens} \in \{26, 99, 194, 435\}$ (see Fig. 1, panel B). To generate longer intervening text samples, we continued the narrative established by the initial preface string ("Before the meeting, Mary wrote down the following list of words:"). For complete texts, see

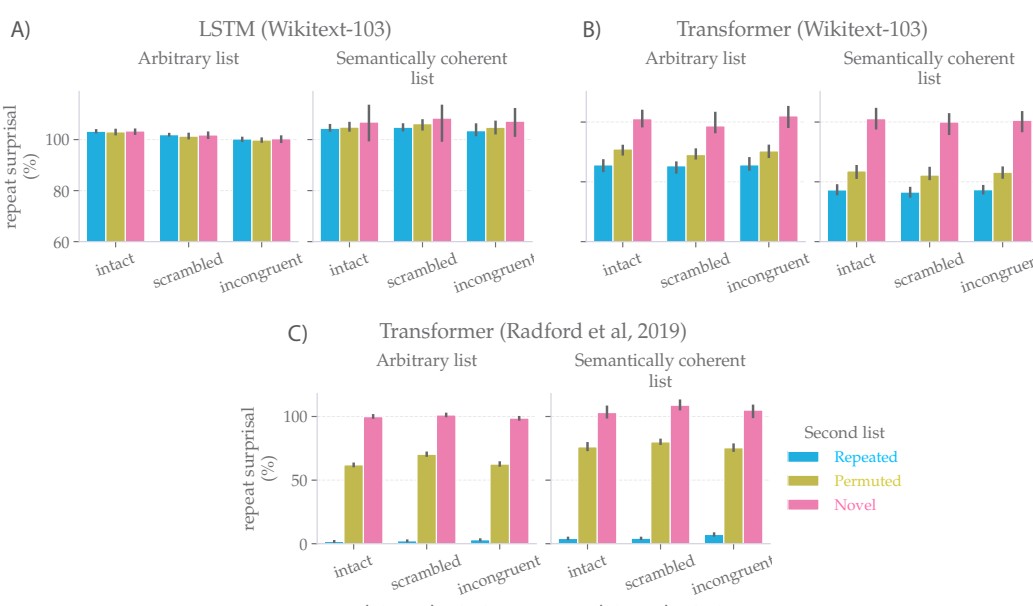

Figure 4: LM memory retrieval for different intervening texts. We plot relative list-median surprisal over all non-initial tokens in lists. Points show group median (over $N^{list} = 230$). Error bars denote 95% confidence interval around the median (bootstrap estimate). Note that in the top-row plots y-axis starts at 60%.

Section A.2 in Appendix A. All intervening text strings ended with the same prompt string ("When she got back, she read the list again:") which introduced the second list.

Memory retrieval in the transformer models, whether trained on Wikitext-103 or a much larger corpus size, was largely invariant to the size of the intervening text between the first and second lists (Fig. 3, B and C, respectively). The results suggest that the two transformers were retrieving prior nouns using a form of direct indexing of the relevant words from the input buffer, rather than implementing a generic memory heuristic, such as predicting that the nouns that have occurred in the most recent 20 tokens will recur.

The results in previous sections suggest that increasing the length of *well-formed, semantically coherent* intervening text does not interfere with memory retrieval in the transformer. In models of human memory, current context, such as immediately preceding text, can indeed be used as a cue for recalling the encoded items (Kahana, 2020). Does the transformers' capacity to retrieve copies of past nouns rely on the content and structure of the intervening text? We tested this by creating incongruent and scrambled versions of the original intervening text (435 tokens). An incongruent condition was created by using intervening text that was syntactically well-formed but semantically incongruent with respect to the preface (see Appendix A.2.2). The scrambled version was created by randomly permuting the tokens of the intervening text.

The transformers' retrieval of past tokens was largely unaffected by the specific content of the intervening text, as long as the intervening text was coherent/well-formed (Fig. 4). However, in GPT-2, median surprisal across permuted arbitrary lists of nouns increased by $\sim 8\%$ when the intervening text was scrambled (Fig. 4, bottom) compared to well-formed text. This suggests that GPT-2 relied on narrative coherence of the intervening text, rather than its aggregate semantic content alone, as a cue for retrieving the ordering information of arbitrary word lists.

### 4.4 TRANSFORMER VERBATIM RECALL IS LEARNED, GUIDED BY ATTENTION, AND REQUIRES DEPTH

Having shown that the transformer LMs could flexibly and robustly retrieve words and their ordering verbatim from short-term memory (Figs. 3 and 4), we next asked: is this ability learned, or does it derive directly from the architecture? To address this question, we re-ran the experiment with varying number of tokens in lists with a randomly initialized transformer model (architecture as in Section 3.3). This random-weights model was unable to retrieve words or their order: surprisal remained at $\sim 100\%$ relative to first lists regardless of whether or not the nouns in the second list have appeared before (Fig. 5, left).

Next we tested whether the transformers' ability to recall past tokens depended on the attention mechanism (Bahdanau et al., 2014; Vaswani et al., 2017) which allows it, in principle, to use all past words, weighted according to their relevance, for next word prediction. The attention function $attn = \texttt{softmax}(\frac{\boldsymbol{QK}^T}{\sqrt{d_k}})\boldsymbol{V}$ is supported by the query and key matrices $\boldsymbol{Q}, \boldsymbol{K} \in \mathbb{R}^{768 \times 768}$. To test for the role of attention in verbatim retrieval, we randomly permuted the rows and columns in each of the 12 attention layers of GPT-2 and reran the experiment with varying number of tokens in lists. The shuffled-attention model retained some capacity to retrieve past nouns (Fig. 5, right), but the effect was greatly reduced (surprisal reduction of $\sim 10\%$ for shuffled-attention, compared with $\sim 100\%$ for the intact model). Intriguingly, this shuffled-attention model showed the same surprisal for repeated and permuted lists, indicating that it was no longer accessing word order information from the original list. Thus, the attention mechanism is necessary for transformers to index past nouns and their order from memory.

Finally, a deep layered architecture is a key characteristic of transformers and performance typically scales with model size (Radford et al., 2019; Kaplan et al., 2020). Does the capacity to perform verbatim recall depend on model depth? To address this question, we trained 1-, 3-, 6- and 12-layer transformers on our 40-million subset of Wikitext-103. Consistent with the hypothesis that depth is crucial, the smaller 1- and 3-layer models show a modest verbatim recall capacity, but were not sensitive to order ($\sim 90\%$ repeat surprisal for repeated and permuted lists, Fig. 7 in A.7). Sensitivity to order progressively emerged in 6- and 12-layer models, where relative surprisal levels were $\sim 5$–$10\%$ lower for repeated relative to permuted lists (Fig. 7 in A.7).

## 5 LSTM RESULTS

### 5.1 THE LSTM RETRIEVES GIST-LIKE MEMORIES OVER SHORT INTERVENING DISTANCES, FACILITATED BY SEMANTIC COHERENCE

The LSTM generated the expectation that the nouns in the second list should belong to the same semantic category and this was weighted towards earlier nouns in the first list. We established this using the same paradigms as described in preceding sections to test memory retrieval capacities of an LSTM LM. If the intervening text was no longer than 26 tokens, LSTM repeat surprisal across non-initial token positions (Figure 3, A, right) revealed a modest decrease ($\sim 5\%$) relative to first list, but only when the nouns in the first and second lists were from the same semantic category. Examining surprisal values broken down by token position in the list (Fig. 2) shows that in semantically coherent lists of nouns, surprisal was higher for novel lists than for repeated or permuted lists, but this memory effect was only present for tokens near the beginning of the list.

In light of this evidence for retrieval in the LSTM across 26 intervening tokens, we examined whether the LSTM retrieves more successfully over much shorter intervals, by reducing the intervening text to 4 tokens of coherent text (see Appendix A.2.4). In this short-range retrieval setting, we now observed a small reduction of relative repeat surprisal of $\sim 5\%$ for arbitrary lists of 3 or 5 nouns as well as a stronger $\sim 15\%$ reduction for semantically coherent lists (Figure 6 in A.6). Overall, the reduction in surprisal was comparable for repeated and permuted lists, indicating that the LSTM did not predict that words would occur in their original order. We did not observe a change in retrieval even with an LSTM that was trained on twice as many tokens and had an approximately 5-fold increase in parameter count (see Fig. 6 in A.6) Taken together, the experiments described in the section suggest that the LSTM retrieves a semantic gist of the prior list. Consistent with this notion of an aggregate

semantic memory, we found that retrieval was stronger for semantically coherent lists, for which an aggregated semantic representation would be closer to each of the individual words in the list.

## 6 DISCUSSION

Short-term memory—the capacity to temporarily store and access recent context for current processing—is a crucial component of language prediction. In this paper, we introduced a paradigm for characterizing a language model's short-term memory capabilities, based on retrieval of verbatim content (sequences of nouns) from prior context, and used this paradigm to analyze LMs with transformer and LSTM architectures.

The transformers we tested were able to access verbatim information – individual tokens and their order – from past context. Furthermore, this verbatim retrieval was learned and largely *resilient* to interference from intervening context. This indicates that the models (especially those trained on the largest corpora) implemented, via learning, a high-resolution memory system. The ability to access individual tokens may in turn support functions that rely on token indexing, akin to the functionality of a general-purpose working memory (WM) buffer proposed in cognitive science (Baddeley, 2003). On the one hand, such flexible WM could subserve the reported ability of transformers to rapidly generalize to new tasks (Brown et al., 2020). Indeed, past work suggests that such meta-learning in traditional RNNs requires a dynamic short-term memory mechanism known as fast weights (Schmidhuber, 1992; Ba et al., 2016) which can be thought of as analogous to self-attention in transformers (Schlag et al., 2021). On the other hand, a highly resilient verbatim memory system could also be disadvantageous if it causes the LM to place too much confidence on verbatim features of prior context for next-word prediction. Indeed, text generated from a transformer LM's predictions can be highly repetitive (Holtzman et al., 2020) – it is possible that an over-reliance on accessing short-term memory may underlie this tendency.

In contrast to the transformers, the LSTM model (in spite of it's larger parameter count) only retrieved a coarse semantic category of previous lists, without fine-grained information about word order, and was only able to do so when the intervening text was short. The tendency of LSTMs to rely on the fuzzy representation of past context for next-word prediction has been reported previously (Khandelwal et al., 2018). Whereas in sequence-to-sequence tasks requiring recall of short lists of pseudowords, recurrent neural networks are a good model of human short-term memory (Botvinick & Plaut, 2006), the copying capacity of LSTMs did not generalize to longer sequences of symbols (Grefenstette et al., 2015). Is tracking a shallow representation of context always a limitation? Not necessarily. Humans frequently maintain a "good-enough" (i.e. gist-like) representation of context (Ferreira & Patson, 2007). When the potential for memory capacity is limited (e.g. when context must be compressed to a single hidden state as in an RNN) maintaining a broad, gist-like – as opposed to token-specific – memory of context may be more *efficient* overall.

Because the transformer models were better overall LMs than the LSTM models (i.e. they exhibited lower perplexity), future work must replicate these results on more powerful LSTM LMs. For example, future work should test LSTMs augmented with attention (Bahdanau et al., 2014) or the copy-mechanism (Gu et al., 2016) or trained with more sophisticated regularization (Merity et al., 2017). That said, it remains challenging to discern which aspects of a model need to be matched across architectures in order to make the comparison fair: while matching corpora is fairly straightforward, it is not clear whether the number of layers or parameters has a common meaning across such distinct architectures.

## 7 CONCLUSIONS

Pretrained language models, and self-supervised predictive learning broadly, have received increased attention in terms of their (in)sufficiency as a framework for achieving feats of human-like language processing (Kaplan et al., 2020; Linzen & Baroni, 2020). Here, we tested the ability of language models to perform an important aspect of human intelligence for natural language — flexibly accessing items from short-term memory — and show that the transformer model, even though not trained with a short-term memory objective, retrieved remarkably detailed representations of past context for predicting new input.

## ETHICS STATEMENT

The research reported here addresses a specific, basic research question about the cognitive organization of short-term memory in contemporary language processing algorithms. Although from a broader perspective, the nature of memory is a key question in developing human-like artificial intelligence, it is, in our opinion, unlikely that the results reported here could pose novel societal challenges as we are primarily trying to better the understanding of the already developed systems.

## REPRODUCIBILITY STATEMENT

Anonymized repository hosting the code used in current experiments is available at the following link: `https://anonymous.4open.science/r/short-term-memory-in-neural-lms-31F9`. The codebase in the repository itself is not self-contained, however, we used the popular scientific python and NLP libraries — any such external dependencies with cursory documentation (including installation) are clearly indicated and provided in the repository's README.md.

All our scripts were used on a high-performance computing (HPC) cluster. Nvidia Quadro RTX 8000 GPUs were used for transformer training. For a single job, a single GPU device was used and a single job typically required ∼6 hours of core-walltime (12-layer model). We tended to request at least 44GB of RAM per job (12-layer model). For a single transformer evaluation, we typically required ∼13:30 (hh:mm) of core-walltime and ∼ 4GB RAM per GPU job. A single LSTM evaluation job typically required ∼ 06:00 (hh:mm) or less of core-walltime and ∼ 8GB RAM per GPU job and these parameter values were sufficient to prevent overflows.

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

# A APPENDIX

## A.1 CODE AND DATA AVAILABILITY

The data will be shared on the project OSF repository: [TBA].

Code with documentation is available at the following anonymized Github repository: https://anonymous.4open.science/r/short-term-memory-in-neural-lms-31F9

## A.2 VIGNETTES

### A.2.1 INTACT

Before the meeting, Mary wrote down the following list of words:

$W_1, W_2, ..., W_N$

$intervening\_text_1$: After the meeting, she took a break and had a cup of coffee. When she got back, she read the list again: $W_1, W_2, ..., W_N$

$intervening\_text_2$: After the meeting, Mary went for a walk. It was a busy day and she needed a break. Outside was really beautiful and warm and the flowers in the park were blooming. When she got back, she read the list again: $W_1, W_2, ..., W_N$

$intervening\_text_3$: After the meeting, Mary went for a walk. It was a busy day and she needed a break. Outside was really beautiful and warm and the flowers in the park were blooming. While she was walking, she listened to the wonderful bird songs. During the walk, Mary could not stop thinking about the meeting. She was thinking about the discussions she had with her coworkers. Luckily, she met her neighbors Sarah and Ryan and they talked briefly. When she got back, she read the list again: $W_1, W_2, ..., W_N$

$intervening\_text_4$: After the meeting, Mary went for a walk. It was a busy day and she needed a break. Outside was really beautiful and warm and the flowers in the park were blooming. While she was walking, she listened to the wonderful bird songs. During the walk, Mary could not stop thinking about the meeting. She was thinking about the discussions she had with her coworkers. Luckily, she met her neighbors Sarah and Ryan and they talked briefly. The couple has just moved to the area from a different city. Mary thought they were very a lovely couple and made good company. They were just getting to know the neighborhood and this was their first time in the park. Mary was curious what were their first impressions of the town. The neighborhood felt very safe to them and they absolutely loved the park. This was only their second time visiting the park. There was so much to discover, so many winding paths and hidden gardens. When she got back, she read the list again: $W_1, W_2, ..., W_N$

$intervening\_text_5$: After the meeting, Mary went for a walk. It was a busy day and she needed a break. Outside was really beautiful and warm and the flowers in the park were blooming. While she was walking, she listened to the wonderful bird songs. During the walk, Mary could not stop thinking about the meeting. She was thinking about the discussions she had with her coworkers. Luckily, she met her neighbors Sarah and Ryan and they talked briefly. The couple has just moved to the area from a different city. Mary thought they were very a lovely couple and made good company. They were just getting to know the neighborhood and this was their first time in the park. Mary was curious what were their first impressions of the town. The neighborhood felt very safe to them and they absolutely loved the park. This was only their second time visiting the park. There was so much to discover, so many winding paths and hidden gardens. It was not a big park by any means, but it offered a quiet refuge where one can escape the worries of everyday life. It also offered opportunities to do sports of all kinds. Young people from around the area played basketball, football, or volleyball. Others took part in outdoor workout sessions. Young families were going on a stroll with their children. Finally, there were so many people who brought their dogs for a walk. It was incredibly satisfying to see the joy our animal friends get when you throw them a ball. All this diversity of people and activities made a walk in this park a truly rewarding and relaxing daily routine. In fact, Sarah and Ryan were thinking of getting a dog. They have not fully decided yet but they really wanted to spend more time outdoors. Mary liked dogs as well, but she was more of a cat person herself. She and her husband had two cats. One was two and the other four years old. They were very independent and spent most of their time outdoors. Mary thought having an animal was a great idea. They talked for a little bit and then Sarah and Ryan invited her to come over for a cup of coffee. Mary said she had time over the weekend. When she got back, she read the list again: $W_1, W_2, ..., W_N$

### A.2.2 INCONGRUENT

Before the meeting, Mary wrote down the following list of words:

$W_1, W_2, ..., W_N$

$intervening\_text_1$: There is a voice in the waters of the great sea. It calls to man continually. When she got back, she read the list again: $W_1, W_2, ..., W_N$

$intervening\_text_2$: Sometimes it thunders in the tempest, when the waves leap high and strong and the wild winds shriek and roar. Sometimes it whispers in the calm, small voice, as if to solicit our regard. When she got back, she read the list again: $W_1, W_2, ..., W_N$

$intervening\_text_3$: After the meeting, Mary went for a walk. It was a busy day and she needed a break. Outside was really beautiful and warm and the flowers in the park were blooming. The sea has much to say; far more than could possibly be comprehended in one volume, however large. It tells us of the doings of man on its broad bosom, from the day in which he first ventured to paddle along shore to the day when he launched his great iron ship, and rushed out to sea. When she got back, she read the list again: $W_1, W_2, ..., W_N$

$intervening\_text_4$: After the meeting, Mary went for a walk. It was a busy day and she needed a break. Outside was really beautiful and warm and the flowers in the park were blooming. The sea has much to say; far more than could possibly be comprehended in one volume, however large. It tells us of the doings of man on its broad bosom, from the day in which he first ventured to paddle along shore to the day when he launched his great iron ship, and rushed out to sea. Before proceeding to the consideration of the wonders connected with and contained in the sea, we shall treat of the composition of the sea itself and of its extent, depth, and bottom. What is the sea made of? Salt water, is the ready reply that rises naturally to every lip. But to this we add the question, what is salt water? To these queries we give the following reply, which, we doubt not, will rather surprise some of our readers. The salt of the ocean varies considerably in different parts. When she got back, she read the list again: $W_1, W_2, ..., W_N$

$intervening\_text_5$: After the meeting, Mary went for a walk. It was a busy day and she needed a break. Outside was really beautiful and warm and the flowers in the park were blooming. The sea has much to say; far more than could possibly be comprehended in one volume, however large. It tells us of the doings of man on its broad bosom, from the day in which he first ventured to paddle along shore to the day when he launched his great iron ship, and rushed out to sea. Before proceeding to the consideration of the wonders connected with and contained in the sea, we shall treat of the composition of the sea itself and of its extent, depth, and bottom. What is the sea made of? Salt water, is the ready reply that rises naturally to every lip. But to this we add the question, what is salt water? To these queries we give the following reply, which, we doubt not, will rather surprise some of our readers. The salt of the ocean varies considerably in different parts. Near the equator, the great heat carries up a larger proportion of water by evaporation than in the more temperate regions. Thus, as salt is not removed by evaporation, the ocean in the torrid zone is salter than in the temperate or frigid zones. The salts of the sea, and other substances contained in it, are conveyed there by the fresh water streams that pour into it from all the continent of the world Here, as these substances cannot be evaporated, they would accumulate to such a degree as to render the ocean uninhabitable by living creatures.The operations of the ocean are manifold. But we cannot speak of these things without making passing reference to the operations of water, as that wonder-working agent of which the ocean constitutes but a part. Nothing in this world is ever lost or annihilated. As the ocean receives all the water that flows from the land, so it returns that water, fresh and pure, in the shape of vapour, to the skies. where, in the form of clouds, it is conveyed to those parts of the earth where its presence is most needed. After having gladdened the heart of man by driving his mills and causing his food to grow, it finds its way again into the sea: and thus the good work goes on with ceaseless regularity. When she got back, she read the list again: $W_1, W_2, ..., W_N$

The incongruent intervening text was sampled from: "The ocean and its wonders" by R. M. Ballantyne. https://www.gutenberg.org/ebooks/21754

### A.2.3 SCRAMBLED

Before the meeting, Mary wrote down the following list of words:

$W_1, W_2, ..., W_N$

$intervening\_text_1$: After a break, a cup and coffee of had she the took meeting. When she got back, she read the list again: $W_1, W_2, ..., W_N$

$intervening\_text_2$: Outside the the beautiful and park flowers blooming were in and was warm really. After, walk for Mary the a went meeting. It needed busy break she day was a and a. When she got back, she read the list again: $W_1, W_2, ..., W_N$

$intervening\_text_3$: Luckily and and met Sarah they Ryan briefly talked her, neighbors she. Thinking during, stop meeting the not about Mary the could walk. The while walking to songs bird listened wonderful, she she was. After, walk for Mary the a went meeting. Had she about she coworkers her

with the was discussions thinking. Outside the the beautiful and park flowers blooming were in and was warm really. It needed busy break she day was a and a. When she got back, she read the list again: $W_1, W_2, ..., W_N$

$intervening\_text_4$: First they their was neighborhood getting and the in park the this to were time know just. There paths so much, and many gardens hidden winding to was discover so. The while walking to songs bird listened wonderful, she she was. Had she about she coworkers her with the was discussions thinking. From the just area city different the a moved couple to has. The absolutely and very them loved they park the safe neighborhood to felt. Outside the the beautiful and park flowers blooming were in and was warm really. And Mary were couple company good lovely made very thought a they. Luckily and and met Sarah they Ryan briefly talked her, neighbors she. Thinking during, stop meeting the not about Mary the could walk. After, walk for Mary the a went meeting. Their this park visiting second was the time only. Impressions Mary what first town the of were was their curious. It needed busy break she day was a and a. When she got back, she read the list again: $W_1, W_2, ..., W_N$

$intervening\_text_5$: It needed busy break she day was a and a. First they their was neighborhood getting and the in park the this to were time know just. Had she about she coworkers her with the was discussions thinking. Of they independent most outdoors time their and were spent very. Get it friends them our joy satisfying when the throw ball a animal to was you see incredibly. The while walking to songs bird listened wonderful, she she was. Weekend had time Mary said the over she. An Mary idea a animal thought great was having. Mary a she was as but cat of herself person more well liked, dogs. It of opportunities kinds sports to also all do offered. Cats husband had she and two her. They spend they really fully but more to outdoors time wanted decided have yet not. A a and of rewarding park all in made this activities relaxing routine daily truly walk people this and diversity. There paths so much, and many gardens hidden winding to was discover so. Finally dogs who were people for brought walk a their so, many there. Luckily and and met Sarah they Ryan briefly talked her, neighbors she. The absolutely and very them loved they park the safe neighborhood to felt. Outside the the beautiful and park flowers blooming were in and was warm really. Young football basketball around played,, volleyball or the people area from. Their this park visiting second was the time only. To Sarah a a for her they Ryan then invited and cup coffee of over come and little talked bit for. From the just area city different the a moved couple to has. And Mary were couple company good lovely made very thought a they. Young with going children stroll on families their a were. Worries a means escape where a offered but one refuge can it by any it the quiet of life everyday, big was park not. Of in Sarah thinking dog a were getting and fact Ryan,. Thinking during, stop meeting the not about Mary the could walk. After, walk for Mary the a went meeting. And one old four the was years other two. Impressions Mary what first town the of were was their curious. Sessions in outdoor others took workout part. When she got back, she read the list again: $W_1, W_2, ..., W_N$

### A.2.4 SHORT

Before the meeting, Mary wrote down the following lists of words. One was:

$W_1, W_2, ..., W_N$

$intervening\_text_1$: And the other: $W_1, W_2, ..., W_N$

### A.3 NOUN LISTS

Arbitrary and semantic word lists are shown in Tables 1 and 2, respectively.

### A.4 TRANSFORMER TRAINING

All models were trained using the PyTorch (Paszke et al., 2019, v1.6) and HuggingFace (Wolf et al., 2020, v4.6.1) python libraries.

### A.4.1 DATASETS

Models were trained on a  40 million token subset of the Wikitext-103 training set (Merity et al., 2016). To create the training set, we selected the first 40 million tokens in the original training set. We then retokenized the dataset with the trained BPE tokenizer (see A.4.2, below). The original

Table 1: Lists of arbitrary nouns used in present experiments.

|    | list |
|----|------|
| 1  | patience, notion, movie, women, canoe, novel, folly, silver, eagle, center. |
| 2  | pleasure, pattern, leader, culture, worker, master, meadow, writer, apple, costume. |
| 3  | paper, belief, factor, total, comrade, angle, battle, pistol, nothing, riches. |
| 4  | cabin, doorway, candle, parent, monarch, kindness, lover, copy, soldier, kingdom. |
| 5  | future, legend, problem, flavor, prairie, forehead, illness, planet, canvas, chamber. |
| 6  | oven, patient, daughter, bubble, colour, product, echo, pepper, fountain, music. |
| 7  | village, shipping, beauty, football, merit, autumn, lumber, research, resort, rival. |
| 8  | county, muscle, vapor, shepherd, sickness, herald, value, mission, finger, building. |
| 9  | iron, onion, opera, attack, prison, butter, interest, colonel, commerce, beggar. |
| 10 | blanket, marriage, ticket, baby, treasure, event, weakness, cottage, cotton, judgment. |
| 11 | summer, bottom, meaning, campaign, voyage, cannon, helmet, thunder, hatred, stanza. |
| 12 | effort, province, parcel, temple, river, major, meeting, career, bargain, chimney. |
| 13 | acre, fortune, motive, question, service, minute, tiger, author, sorrow, parlor. |
| 14 | motor, lawyer, powder, habit, mountain, district, learning, leather, hero, water. |
| 15 | orange, letter, acid, stocking, olive, garden, feeling, motion, compass, model. |
| 16 | island, theory, person, season, supper, reason, patent, picture, custom, twilight. |
| 17 | dragon, pillow, aspect, chairman, marble, horror, justice, danger, bedroom, canal. |
| 18 | writing, pocket, training, circuit, cousin, chapter, quarter, button, turkey, surface. |
| 19 | sailor, matter, darkness, scatter, captain, tunnel, method, wagon, effect, arrow. |
| 20 | image, butcher, anchor, scholar, compound, tribute, victim, lily, witness, widow. |
| 21 | candy, window, detail, ocean, program, traffic, feather, array, pilot, silence. |
| 22 | vessel, robber, banner, kitten, lemon, failure, princess, painter, bullet, rifle. |
| 23 | engine, timber, harbour, party, level, money, single, system, unit, traitor. |

Table 2: Lists of semantic nouns used in present experiments.

|    | list |
|----|------|
| 1  | window, door, roof, wall, floor, ceiling, room, basement, hearth, hall. |
| 2  | leg, arms, head, eye, foot, nose, finger, ear, hand, toe. |
| 3  | sailboat, destroyer, battleship, cruiser, submarine, yacht, canoe, freighter, tugboat, steamship. |
| 4  | robin, sparrow, heron, eagle, crow, hawk, parrot, pigeon, woodpecker, vulture. |
| 5  | apple, pear, banana, peach, grape, cherry, plum, grapefruit, lemon, apricot. |
| 6  | hammer, saw, nails, level, plane, chisel, ruler, wrench, drill, screws. |
| 7  | hurricane, tornado, rain, snow, hail, storm, wind, cyclone, clouds, sunshine. |
| 8  | oxygen, hydrogen, nitrogen, carbon, sodium, sulphur, helium, chlorine, calcium, potassium. |
| 9  | chemistry, physics, psychology, biology, zoology, botany, astronomy, mathematics, geology, microbiology. |
| 10 | piano, drum, trumpet, violin, clarinet, flute, guitar, saxophone, trombone, oboe. |
| 11 | knife, spoon, fork, pan, pot, stove, bowl, mixer, cup, dish. |
| 12 | trout, shark, herring, perch, salmon, tuna, goldfish, cod, carp, pike. |
| 13 | football, baseball, basketball, tennis, swimming, soccer, golf, hockey, lacrosse, badminton. |
| 14 | doctor, lawyer, teacher, dentist, engineer, professor, carpenter, salesman, nurse, psychologist. |
| 15 | oak, maple, pine, elm, birch, spruce, redwood, walnut, fir, hickory. |
| 16 | shirt, socks, pants, shoes, blouse, skirt, coat, dress, hat, sweater. |
| 17 | cancer, measles, tuberculosis, polio, malaria, leukemia, pneumonia, smallpox, influenza, encephalitis. |
| 18 | mountain, hill, valley, river, rock, lake, canyon, tundra, ocean, cave. |
| 19 | murder, rape, robbery, theft, assault, arson, kidnapping, larceny, adultery, battery. |
| 20 | log, cat, horse, cow, lion, tiger, elephant, pig, bear, mouse. |
| 21 | fly, ant, bee, mosquito, spider, beetle, wasp, moth, flea, butterfly. |
| 22 | blue, red, green, yellow, black, purple, white, pink, brown, blonde. |
| 23 | cotton, wool, silk, rayon, linen, satin, velvet, denim, canvas, felt. |

Table 3: Architecture and model parameters.

|  | 1 layer | 3 layer | 6 layer | 12 layer |
|---|---|---|---|---|
| **activation function** | gelu_new | gelu_new | gelu_new | gelu_new |
| **n layer** | 1 | 3 | 6 | 12 |
| **n head** | 3 | 3 | 6 | 12 |
| **n ctx** | 1024 | 1024 | 1024 | 1024 |
| **n positions** | 1024 | 1024 | 1024 | 1024 |
| **vocab size** | 28439 | 28439 | 28439 | 28439 |
| **per device train batch size** | 12 | 12 | 12 | 12 |
| **per device eval batch size** | 12 | 12 | 12 | 12 |
| **learning rate** | 0.00007 | 0.00007 | 0.00007 | 0.00006 |
| **adam beta1** | 0.6 | 0.6 | 0.6 | 0.6 |
| **adam beta2** | 0.05 | 0.05 | 0.05 | 0.1 |
| **n params (M)** | 29.7 | 43.9 | 65.2 | 107.7 |

Verbatim retrieval of words in randomly initialized and shuffled attention models

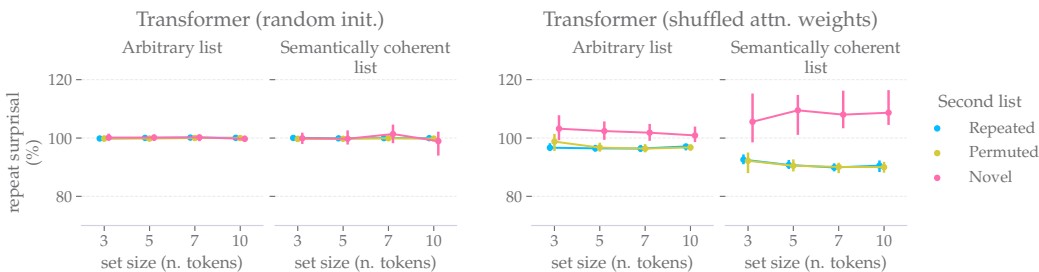

Figure 5: Repeat surprisal for randomly initialized transformer LM and a transformer with permuted attention weights. Reported is relative list-median surprisal over all non-initial tokens in lists only. Points show group median (over $N^{list} = 230$). Error bars denote 95% confidence interval around the median (bootstrap estimate). Note that in these plots y-axis starts at 70%.

validation and test sets were used for model validation and testing, respectively. The retokenized datasets (i.e. indices marking vocabulary items) for the train, validation, and test set are available under the files tab of our OSF repository (see Section A.1).

### A.4.2 TOKENIZER

To control the vocabulary size, we retrained the BPE tokenizer on the concatenated Wikitext-103 training, evaluation and test sets. Vocabulary size was set to 28,439 entries and a minimum token occurrence frequency of 2 was used as inclusion criterion. The resulting `merges.txt` and vocabulary (vocab.json) files are available under the files tab of the project OSF repository[5]. To perform training, we used the `tokenizer.train()` module by HuggingFace (Wolf et al., 2020) library (see `dataset.py`).

### A.5 RANDOMLY INITIALIZED TRANSFORMER, TRANSFORMER WITH SHUFFLED ATTENTION

The results for randomly initialized transformer and the transformer with shuffled attention are shown in Fig. 5.

### A.6 LSTM VERBATIM RECALL OVER 3-TOKEN CONTEXT

The results for LSTM verbatim recall over short context (3 tokens) are shown in Fig. 6.

---

[5]link TBA

Verbatim retrieval of words after intervening text of 3 tokens

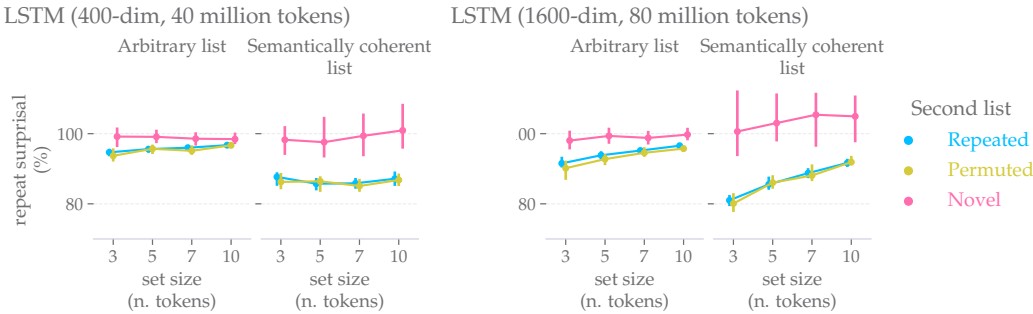

Figure 6: LSTM verbatim token retrieval for varying number of tokens being retrieved at short (3-token) intervening text. Reported is proportion of list-median surprisal on second relative to first list of nouns (repeat surprisal). Points show group median (over $N^{list} = 230$). Error bars denote 95% confidence interval around the median (bootstrap estimate).

Verbatim retrieval of words with increasing transformer depth

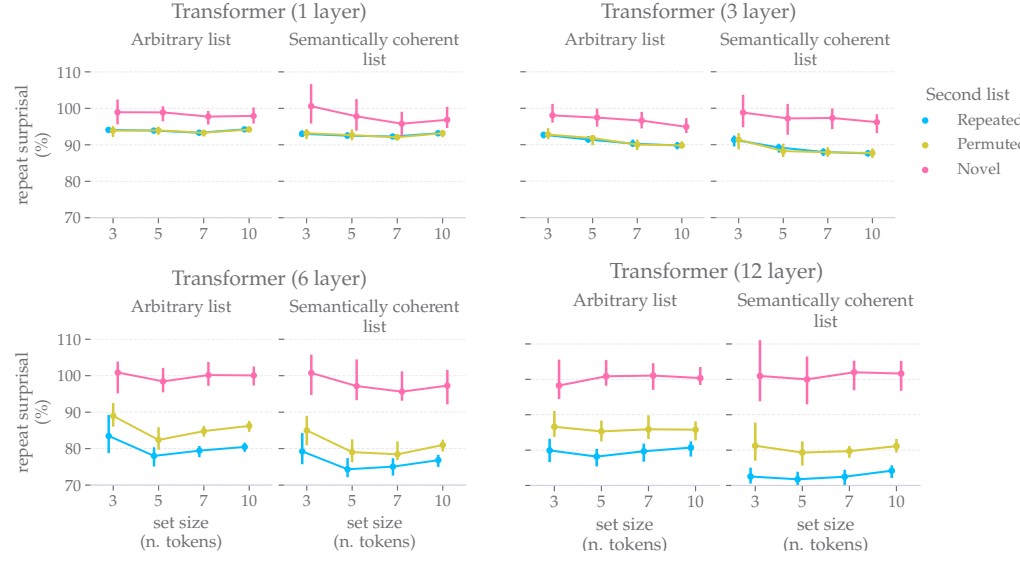

Figure 7: LM memory retrieval for models of different depths. Reported is relative list-median surprisal over all non-initial tokens in lists. Points show group median (over $N^{list} = 230$). Error bars denote 95% confidence interval around the median (bootstrap estimate). Note that in these plots y-axis starts at 70%.

## A.7 VERBATIM RECALL REQUIRES DEPTH

The results for transformers at different number of layers (depth) are shown in Fig. 7.

## A.8 RESULTS ON CONTROL VIGNETTES

The results with additional control vignettes are shown in Figures 8 (Wikitext-103 transformer) and 9 (GPT-2).

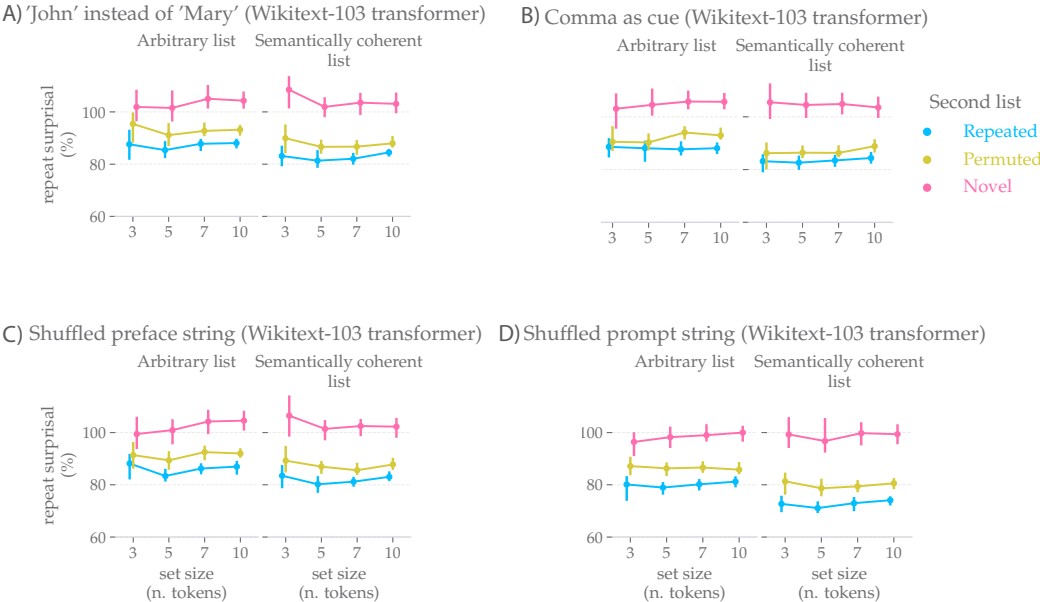

Figure 8: Wikitext-103 transformer memory retrieval results for control vignettes. Top left: in this vignette the subject 'Mary' is replaced with 'John'. Top right: in this vignette the colon introducing the list is replaced with a comma. Bottom: the preface string is randomly permuted. Reported is relative list-median surprisal over all non-initial tokens in lists. Points show group median (over $N^{list}$ = 230). Error bars denote 95% confidence interval around the median (bootstrap estimate). Note that in these plots y-axis starts at 70%.

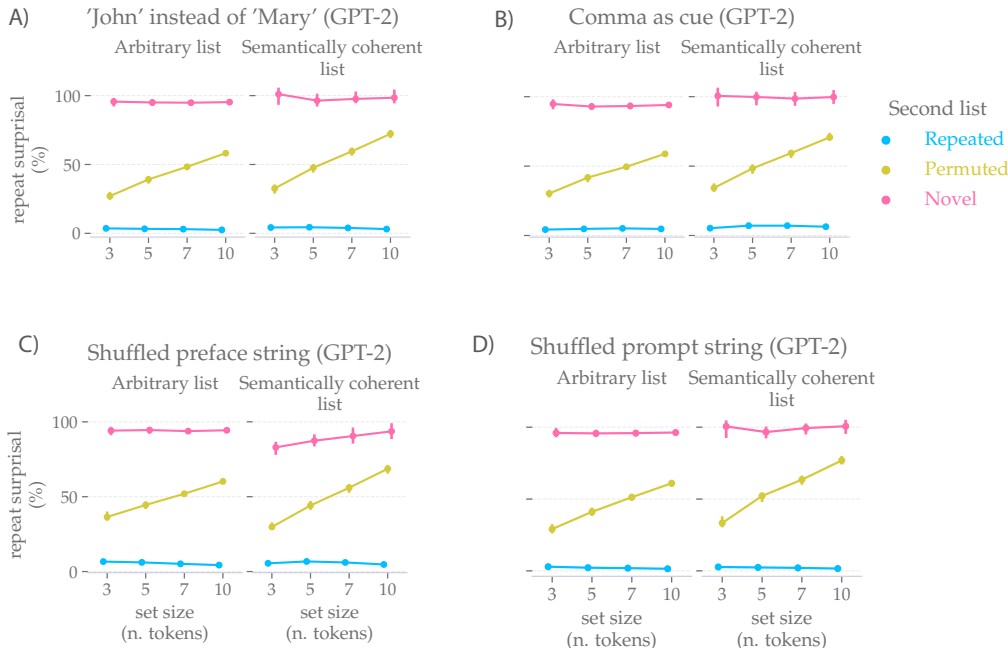

Figure 9: Results for paradigms with additional control vignettes: A) the subject 'Mary' is replaced with 'John', B) the colon token introducing the list is replaced with a comma, C) the tokens in the preface string are randomly permuted and, D) the tokens in the prompt string are randomly permuted. Reported is relative list-median surprisal over all non-initial tokens in lists. Points show group median (over $N^{list}$ = 230). Error bars denote 95% confidence interval around the median (bootstrap estimate).

Analysis using the mean as the estimator of group statistic

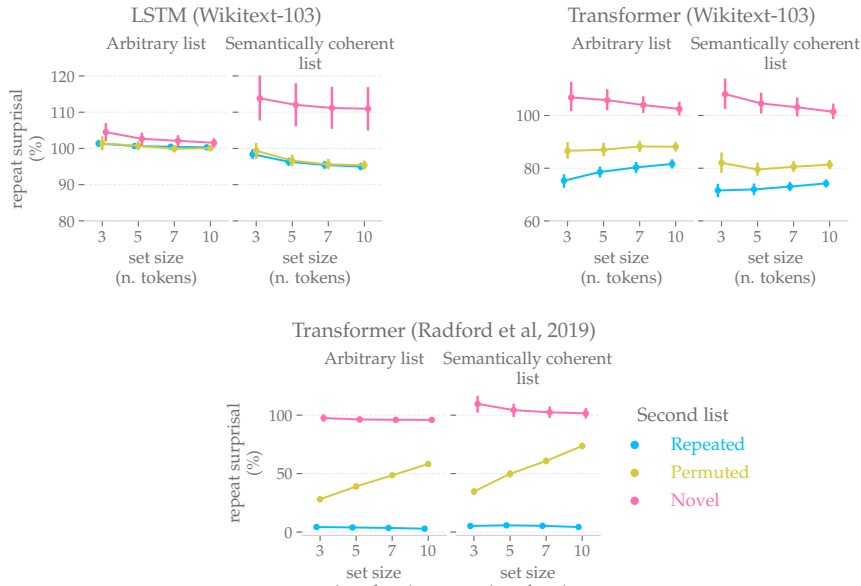

Figure 10: Verbatim token retrieval for varying number of tokens being retrieved (left) and the length of the intervening text (right). Reported is proportion of list-median surprisal on second relative to first list of nouns. Points show group mean (over $N^{list} = 230$). Error bars denote 95% confidence interval around the mean (bootstrap estimate). For set size manipulation, intervening text is fixed at 26 tokens. For intervening text manipulation, set size is fixed at 10 tokens.

Table 4: LSTM-400 (Wikitext-103) surprisal values for four initial token positions, list type and second list condition.

| List | Token position Second list | 0 | 1 | 2 | 3 |
|------|------|---|---|---|---|
| **arbitrary** | **Novel** | 18.5 (18.0-18.8) | 16.1 (15.5-16.5) | 15.6 (15.2-16.0) | 15.5 (15.1-15.9) |
| | **Permuted** | 18.1 (17.5-18.5) | 15.3 (15.0-15.6) | 15.4 (15.1-15.8) | 15.2 (14.9-15.4) |
| | **Repeated** | 18.0 (17.7-18.4) | 15.7 (15.2-16.0) | 15.5 (15.1-16.0) | 15.5 (15.1-15.8) |
| **semantic** | **Novel** | 19.5 (19.1-19.9) | 13.2 (12.7-13.7) | 11.8 (11.4-12.1) | 10.9 (10.5-11.3) |
| | **Permuted** | 17.6 (17.2-17.8) | 12.7 (11.9-13.1) | 9.9 (9.4-10.4) | 9.7 (9.1-10.2) |
| | **Repeated** | 17.5 (17.1-17.8) | 12.0 (11.2-12.5) | 10.5 (9.9-10.9) | 10.0 (9.4-10.5) |

## A.9 USING THE MEAN AS THE ESTIMATOR

Results of the analyses using the mean, rather than median, as the estimator are shown in Fig. 10.

## A.10 RESULTS TABLES

### A.10.1 TABLES FOR SURPRISAL TIME-COURSES (FIGURE 2)

Numerical values for data shown in Fig. 2 are shown in Tables 4 (LSTM), 5 (Wikitext-103 transformer), and 6 (Radford et al. 2019).

### A.10.2 TABLES FOR SET-SIZE MANIPULATION (FIGURE 3)

Numerical values for data shown in Fig. 3 (left) are shown in Tables 7 (LSTM), 8 (Wikitext-103 transformer), and 9 (Radford et al. 2019).

Table 5: Transformer (Wikitext-103) surprisal values for four initial token positions, list type and second list condition.

| List | Token position
Second list | 0 | 1 | 2 | 3 |
|------|------|---|---|---|---|
| **arbitrary** | **Novel** | 12.3 (12.1-12.5) | 12.5 (11.9-13.3) | 11.8 (11.4-12.3) | 12.3 (11.8-13.0) |
| | **Permuted** | 12.0 (11.6-12.2) | 11.6 (11.0-12.3) | 11.3 (11.0-11.7) | 12.1 (11.6-12.6) |
| | **Repeated** | 5.4 (4.2-6.5) | 11.4 (10.9-11.9) | 10.8 (10.3-11.3) | 10.7 (10.1-11.3) |
| **semantic** | **Novel** | 12.8 (12.6-13.6) | 13.0 (12.2-14.1) | 13.2 (12.3-13.9) | 12.6 (12.0-13.4) |
| | **Permuted** | 12.0 (11.6-12.4) | 11.6 (11.2-12.2) | 11.7 (11.2-12.2) | 11.8 (11.4-12.2) |
| | **Repeated** | 5.1 (4.4-6.4) | 11.4 (10.8-12.0) | 11.1 (10.6-11.7) | 11.0 (10.5-11.5) |

Table 6: Transformer (Radford et al, 2019) surprisal values for four initial token positions, list type and second list condition.

| List | Token position
Second list | 0 | 1 | 2 | 3 |
|------|------|---|---|---|---|
| **arbitrary** | **Novel** | 10.4 (10.2-10.6) | 9.6 (9.3-9.8) | 9.4 (9.1-9.6) | 9.0 (8.8-9.2) |
| | **Permuted** | 7.9 (7.5-8.0) | 5.9 (5.6-6.2) | 5.6 (5.2-5.8) | 5.6 (5.4-6.0) |
| | **Repeated** | 1.9 (1.8-2.0) | 0.5 (0.5-0.6) | 0.3 (0.3-0.3) | 0.2 (0.2-0.2) |
| **semantic** | **Novel** | 11.5 (11.1-11.9) | 5.9 (5.5-6.1) | 5.0 (4.7-5.5) | 4.7 (4.4-4.9) |
| | **Permuted** | 5.2 (4.9-5.5) | 3.7 (3.5-4.0) | 3.3 (3.1-3.5) | 3.3 (3.0-3.4) |
| | **Repeated** | 1.9 (1.8-2.0) | 0.3 (0.3-0.3) | 0.2 (0.2-0.3) | 0.2 (0.2-0.2) |

Table 7: LSTM (Wikitext-103) word list surprisal as a function of set size. We report the percentage of list-median surprisal on second relative to first lists. Ranges are 95% confidence intervals around the observed median (bootstrap estimate, $N^{resample} = 1000$). The length of intervening text is fixed at 26 tokens.

| List | Set-Size
Condition | 3 | 5 | 7 | 10 |
|------|------|---|---|---|---|
| **Arbitrary** | **Novel** | 103.4% (100.1-105.9) | 102.6% (100.7-104.7) | 101.8% (100.4-103.0) | 101.3% (99.9-102.1) |
| | **Permuted** | 101.2% (99.4-102.5) | 100.7% (99.9-101.8) | 99.2% (98.7-100.8) | 100.1% (99.6-100.5) |
| | **Repeated** | 101.1% (100.6-102.1) | 100.8% (100.2-101.3) | 100.6% (100.1-101.0) | 100.4% (100.0-100.7) |
| **Semantic** | **Novel** | 103.2% (98.4-108.4) | 102.1% (97.5-108.7) | 102.5% (96.5-109.5) | 107.1% (95.0-110.7) |
| | **Permuted** | 98.4% (95.0-101.2) | 96.1% (94.5-97.8) | 95.1% (93.5-96.7) | 95.1% (94.3-96.2) |
| | **Repeated** | 98.1% (96.6-99.2) | 95.8% (95.4-97.0) | 95.6% (95.1-96.3) | 95.2% (94.6-96.1) |

Table 8: Transformer (Wikitext-103) word list surprisal as a function of set size. We report the percentage of list-median surprisal on second relative to first lists. Ranges are 95% confidence intervals around the observed median (bootstrap estimate, $N^{resample} = 1000$). The length of intervening text is fixed at 26 tokens.

| List | Set-Size
Condition | 3 | 5 | 7 | 10 |
|------|------|---|---|---|---|
| **Arbitrary** | **Novel** | 96.3% (93.5-100.7) | 100.4% (95.5-103.7) | 100.7% (97.9-105.2) | 101.4% (97.8-104.0) |
| | **Permuted** | 84.8% (82.7-88.7) | 86.0% (83.7-88.0) | 87.0% (85.3-89.5) | 86.8% (84.7-89.2) |
| | **Repeated** | 75.0% (71.4-79.1) | 78.1% (76.4-80.5) | 79.7% (77.2-82.1) | 81.3% (79.4-83.4) |
| **Semantic** | **Novel** | 100.0% (94.0-107.3) | 97.5% (94.3-106.1) | 100.5% (97.1-104.2) | 101.3% (97.3-103.4) |
| | **Permuted** | 79.9% (76.4-83.6) | 78.4% (76.2-82.0) | 80.6% (78.0-81.9) | 81.2% (80.2-82.6) |
| | **Repeated** | 71.4% (69.8-73.8) | 70.4% (68.9-72.8) | 72.2% (70.6-74.5) | 74.7% (73.1-75.9) |

Table 9: Transformer (Radford et al, 2019) word list surprisal as a function of set size. We report the percentage of list-median surprisal on second relative to first lists. Ranges are 95% confidence intervals around the observed median (bootstrap estimate, $N^{resample} = 1000$). The length of intervening text is fixed at 26 tokens.

| List | Set-Size Condition | 3 | 5 | 7 | 10 |
|------|------|------|------|------|------|
| **Arbitrary** | **Novel** | 96.3% (93.9-97.9) | 95.3% (93.4-96.9) | 95.1% (94.4-96.1) | 95.7% (94.7-96.6) |
| | **Permuted** | 27.2% (25.4-29.5) | 39.0% (37.6-41.0) | 48.3% (47.3-49.3) | 58.2% (57.2-59.4) |
| | **Repeated** | 3.8% (3.4-4.2) | 3.6% (3.3-3.9) | 3.3% (2.9-3.4) | 2.7% (2.5-2.9) |
| **Semantic** | **Novel** | 102.0% (94.8-105.0) | 97.9% (92.5-101.0) | 99.1% (95.2-102.0) | 99.6% (96.0-103.8) |
| | **Permuted** | 32.6% (30.1-36.2) | 47.7% (45.1-49.5) | 59.9% (57.7-61.7) | 72.4% (70.7-74.2) |
| | **Repeated** | 4.5% (4.2-4.7) | 4.6% (4.2-5.1) | 4.1% (3.7-4.8) | 3.3% (3.0-3.6) |

Table 10: LSTM (Wikitext-103) word list surprisal as a function of intervening text size. We report the percentage of list-median surprisal on second relative to first lists. Ranges are 95% confidence intervals around the observed median (bootstrap estimate, $N^{resample} = 1000$). The list length is fixed at 10 tokens.

| List | Intervening text len. Condition | 26 | 99 | 194 | 435 |
|------|------|------|------|------|------|
| **Arbitrary** | **Novel** | 101.3% (99.9-102.1) | 103.2% (102.5-104.1) | 101.2% (99.7-102.1) | 103.4% (102.3-103.9) |
| | **Permuted** | 100.1% (99.6-100.5) | 102.9% (102.1-103.7) | 100.8% (100.2-101.5) | 103.0% (102.1-103.8) |
| | **Repeated** | 100.4% (100.0-100.7) | 103.2% (102.8-103.6) | 101.2% (100.8-101.5) | 103.2% (102.9-103.6) |
| **Semantic** | **Novel** | 107.1% (95.0-110.7) | 107.7% (99.7-111.6) | 101.0% (94.0-108.9) | 106.9% (99.8-113.2) |
| | **Permuted** | 95.1% (94.3-96.2) | 102.1% (100.7-102.6) | 100.3% (99.3-101.1) | 104.9% (103.6-106.5) |
| | **Repeated** | 95.2% (94.6-96.1) | 101.5% (100.8-102.4) | 99.4% (98.3-100.3) | 104.4% (103.6-105.6) |

### A.10.3 Tables for intervening text length manipulation (Figure 3)

Numerical values for data shown in Fig. 3 (right) are shown in Tables 10 (LSTM), 11 (Wikitext-103 transformer), and 12 (Radford et al. 2019).

### A.10.4 Numerical data for transformer depth results (Figure 7)

Numerical values for data 1-, 3-, 6-, and 12-layer transformer (Fig. 7) are shown in Tables 13, 14, 15, and 16, respectively.

### A.11 Compute resources

Nvidia K80 GPUs were used to run jobs involved in GPT-2 evaluation. For a single job, a single GPU device was used and a single job typically required ~12 hours of core-walltime and $\sim 1.5$ GB of RAM. We tended to request at least 13:00 (hh:mm) of walltime and 5GB of RAM per job.

Table 11: Transformer (Wikitext-103) word list surprisal as a function of intervening text size. We report the percentage of list-median surprisal on second relative to first lists. Ranges are 95% confidence intervals around the observed median (bootstrap estimate, $N^{resample} = 1000$). The list length is fixed at 10 tokens.

| List | Intervening text len. Condition | 26 | 99 | 194 | 435 |
|------|------|------|------|------|------|
| **Arbitrary** | **Novel** | 101.4% (97.8-104.0) | 100.7% (98.0-103.9) | 101.7% (98.8-105.1) | 101.1% (98.6-103.7) |
| | **Permuted** | 86.8% (84.7-89.2) | 87.6% (86.1-89.1) | 90.4% (88.7-91.9) | 91.0% (89.2-92.0) |
| | **Repeated** | 81.3% (79.4-83.4) | 81.7% (79.9-83.7) | 84.5% (82.8-86.3) | 85.7% (83.7-87.1) |
| **Semantic** | **Novel** | 101.3% (97.3-103.4) | 100.7% (96.6-103.3) | 101.5% (97.6-105.0) | 101.1% (97.9-104.3) |
| | **Permuted** | 81.2% (80.2-82.6) | 81.0% (79.8-83.2) | 83.2% (81.7-84.6) | 83.7% (81.4-85.2) |
| | **Repeated** | 74.7% (73.1-75.9) | 74.4% (72.4-75.9) | 76.5% (74.7-78.4) | 77.3% (75.9-78.8) |

Table 12: Transformer (Radford et al, 2019) word list surprisal as a function of intervening text size. We report the percentage of list-median surprisal on second relative to first lists. Ranges are 95% confidence intervals around the observed median (bootstrap estimate, $N^{resample} = 1000$). The list length is fixed at 10 tokens.

| List | Intervening text len. Condition | 26 | 99 | 194 | 435 |
|---|---|---|---|---|---|
| Arbitrary | Novel | 95.7% (94.7-96.6) | 97.5% (96.7-98.4) | 98.3% (97.3-98.9) | 99.9% (99.2-100.9) |
| | Permuted | 58.2% (57.2-59.4) | 59.7% (58.8-60.6) | 61.8% (60.6-63.1) | 62.0% (61.0-62.8) |
| | Repeated | 2.7% (2.5-2.9) | 1.7% (1.6-1.8) | 1.7% (1.5-1.8) | 1.8% (1.6-1.9) |
| Semantic | Novel | 99.6% (96.0-103.8) | 101.7% (97.2-104.9) | 103.5% (98.1-105.7) | 103.1% (99.4-107.6) |
| | Permuted | 72.4% (70.7-74.2) | 73.7% (72.1-75.1) | 75.0% (73.3-77.2) | 76.1% (73.8-78.9) |
| | Repeated | 3.3% (3.0-3.6) | 2.8% (2.6-3.1) | 2.9% (2.6-3.2) | 4.1% (3.8-4.6) |

Table 13: Transformer (1 layer) word list surprisal as a function of set size. We report the percentage of list-median surprisal on second relative to first lists. Ranges are 95% confidence intervals around the observed median (bootstrap estimate ($N^{resample} = 1000$).

| List | Set-Size Condition | 3 | 5 | 7 | 10 |
|---|---|---|---|---|---|
| Arbitrary | Novel | 98.9% (96.0-102.1) | 98.9% (96.8-100.2) | 97.7% (95.9-99.0) | 97.9% (96.2-99.9) |
| | Permuted | 93.9% (92.5-94.8) | 94.0% (93.1-94.7) | 93.3% (92.6-93.7) | 94.2% (93.8-94.7) |
| | Repeated | 94.1% (93.6-94.5) | 93.9% (93.7-94.2) | 93.3% (93.0-93.8) | 94.2% (94.1-94.4) |
| Semantic | Novel | 100.6% (96.2-106.3) | 97.8% (94.2-102.2) | 95.8% (92.8-98.7) | 96.9% (95.0-100.1) |
| | Permuted | 93.2% (91.9-94.0) | 92.7% (91.6-93.9) | 92.1% (91.5-92.6) | 93.2% (92.4-93.7) |
| | Repeated | 93.0% (92.6-93.6) | 92.5% (92.1-93.1) | 92.3% (92.0-92.7) | 93.2% (92.9-93.6) |

Table 14: Transformer (3 layer) word list surprisal as a function of set size. We report the percentage of list-median surprisal on second relative to first lists. Ranges are 95% confidence intervals around the observed median (bootstrap estimate ($N^{resample} = 1000$).

| List | Set-Size Condition | 3 | 5 | 7 | 10 |
|---|---|---|---|---|---|
| Arbitrary | Novel | 98.0% (96.4-100.8) | 97.4% (95.3-99.6) | 96.6% (94.8-98.7) | 94.9% (93.6-96.9) |
| | Permuted | 92.8% (91.8-94.2) | 91.8% (90.2-92.3) | 90.0% (88.9-91.1) | 89.8% (89.3-90.6) |
| | Repeated | 92.7% (92.1-93.2) | 91.4% (90.9-91.9) | 90.3% (89.7-91.1) | 89.8% (89.1-90.4) |
| Semantic | Novel | 98.8% (95.1-103.4) | 97.2% (93.1-100.8) | 97.3% (94.6-99.6) | 96.2% (93.5-98.0) |
| | Permuted | 91.4% (89.0-92.9) | 88.3% (87.0-89.9) | 87.9% (87.0-89.0) | 87.7% (86.7-88.6) |
| | Repeated | 91.4% (89.8-92.2) | 89.2% (88.2-89.9) | 88.0% (87.2-88.7) | 87.6% (87.2-88.2) |

Table 15: Transformer (6 layer) word list surprisal as a function of set size. We report the percentage of list-median surprisal on second relative to first lists. Ranges are 95% confidence intervals around the observed median (bootstrap estimate ($N^{resample} = 1000$).

| List | Set-Size Condition | 3 | 5 | 7 | 10 |
|---|---|---|---|---|---|
| Arbitrary | Novel | 100.9% (95.5-103.5) | 98.4% (95.8-101.8) | 100.2% (97.6-103.4) | 100.1% (97.6-102.2) |
| | Permuted | 89.0% (86.4-92.1) | 82.4% (80.0-85.5) | 84.8% (83.6-85.9) | 86.2% (85.0-87.3) |
| | Repeated | 83.5% (79.1-88.9) | 78.0% (75.4-80.0) | 79.4% (78.0-80.4) | 80.5% (79.4-81.3) |
| Semantic | Novel | 100.7% (95.0-105.4) | 97.1% (93.6-104.1) | 95.6% (93.5-100.9) | 97.3% (92.5-101.3) |
| | Permuted | 85.0% (81.3-88.6) | 79.0% (76.5-82.2) | 78.4% (77.1-81.6) | 81.0% (79.6-82.1) |
| | Repeated | 79.2% (76.0-83.9) | 74.3% (72.5-77.1) | 75.0% (72.9-77.0) | 76.8% (75.3-78.0) |

Table 16: Transformer (12 layer) word list surprisal as a function of set size. We report the percentage of list-median surprisal on second relative to first lists. Ranges are 95% confidence intervals around the observed median (bootstrap estimate ($N^{resample} = 1000$).

| List | Set-Size Condition | 3 | 5 | 7 | 10 |
|------|--------------------|---|---|---|-----|
| **Arbitrary** | **Novel** | 98.3% (94.8-105.2) | 100.9% (98.6-105.1) | 101.1% (97.4-104.2) | 100.4% (98.8-103.1) |
| | **Permuted** | 86.6% (84.1-90.7) | 85.3% (82.8-88.0) | 85.9% (83.5-89.5) | 85.8% (83.1-87.7) |
| | **Repeated** | 79.9% (76.9-82.8) | 78.2% (75.7-80.0) | 79.7% (77.1-81.5) | 80.7% (78.5-82.1) |
| **Semantic** | **Novel** | 101.0% (94.2-110.7) | 100.0% (93.3-106.1) | 102.0% (97.3-104.9) | 101.6% (97.1-104.8) |
| | **Permuted** | 81.2% (77.4-87.4) | 79.3% (76.0-82.1) | 79.7% (78.2-80.8) | 81.1% (79.7-82.8) |
| | **Repeated** | 72.5% (70.9-74.7) | 71.7% (69.6-73.5) | 72.5% (70.3-74.1) | 74.2% (72.5-75.3) |

CPUs were use to evaluate RNN models. A single job typically required ~04:30 (hh:mm) of core-walltime and ~ 1.5GB RAM per job. To our experience requesting 06:00 (hh:mm) of walltime and 4GB is typically sufficient to avoid overflows.

