# OpenReview forum: "Short-term memory in neural language models"
_ICLR.cc/2022/Conference — ICLR 2022 Submitted_

### Official Review · Reviewer_ax86 · 2021-10-25

**Correctness:** 3
**Technical Novelty And Significance:** 2
**Empirical Novelty And Significance:** 2
**Recommendation:** 5
**Confidence:** 4

**Main Review:**

The paper proposes a task for pretrained language models: extract in verbatim a list of nouns occurred in previous context. The authors compare two families of models (i.e. transformers and LSTMs) using the constructed benchmark and show some fruitful results. The main conclusion seems to be that transformers can perform robust verbatim recall while LSTM tends to store a semantic gist. The other interesting results include that transformer performance on this task improves with corpus size and transformer depth, and the performance of transformers is linked to the attention patterns.

However, the paper doesn’t provide a deep dive into the topics (e.g. why pretrained transformers excel at this exact extraction task), although the authors propose some research directions in general in the conclusion. In my humble opinion, there are at least two directions that might give insights to explain such superior performance for transformers (which I think also are related works to the paper that should be appropriately discussed, the third one might be remotely related):

#Attention
Attention can be integrated into recurrent neural networks (https://arxiv.org/abs/1409.0473). If attention is the key ingredient that makes transformers perform better at this task, it will be interesting to see if integrating attention in RNNs can help with its performance.

#Copy mechanism
The copy mechanism is a standard architecture for neural autoregressive models (see pointer net https://arxiv.org/pdf/1506.03134.pdf and copy mechanism https://arxiv.org/abs/1603.06393 as examples). Different questions that can be asked such as how copy (in RNNs and transformers) can help this task; will transformers continue to benefit from having a copy mechanism? What is the difference between transformer only and transformer + copy mechanism etc.

#Linguistic structures (optional)
The robustness of transformers to intervening text might be attributed to transformers' capability of getting the linguistic instruction to copy, maybe particular for large models like GPT-2 (https://nlp.stanford.edu/pubs/hewitt2019structural.pdf). The experiments don’t seem to support or argue against such possible explanations while this might be interesting and relevant.

In conclusion, I think the constructed benchmark sheds some light on the inner workings of transformers; but these results are not well discussed with existing literature, making the novelty hard to spot and the presented research look not deep enough for ICLR.

Minor writing/clarification:
- In section 4.3 final paragraph, I suggest adding "absolute value" for the part involving GPT-2 increased 8%.
- In discussion, it is said that "overly effective short-term memory may underlie the tendency for transformers to repeat" and cite (Holtzman et al. 2020). I don't think such argument is put forward in (Holtzman et al. 2020) which mainly discusses sampling strategy to improve generation performance (which mitigates the repetition issue).
- In discussion, "Similar limitations of LSTMs in retrieving past context have been shown in past work", I think readers expect some references at the end of this sentence (some of the references afterwards can fit here).



**Summary Of The Paper:**

Leveraging the verbatim extraction task, the authors compare LSTM and transformer architectures showing transfomers perform robust verbatim recall while LSTM tends to store a semantic gist; the question of depth, attention and how long such memory are stored are also investigated.

**Summary Of The Review:**

By using verbatim extraction as a task, the authors reveal the difference of two popular autoregressive neural architectures: LSTM and Transformers. The authors also show that the phenomenon is related to the attention pattern, and have answered different questions including model depth, training size, intervening text impacts on the results.

However, many related works do not seem to be well compared or even discussed. These include at least attention in RNNs and copy mechanism, and maybe also related to linguistic structures that transformer might capture. These missings make some important research questions unanswered for this work.

---

> ### Author Response · Authors · 2021-11-22
> **Causal LMs are better motivated from the standpoint of human processing phenomena; additional architectural features would likely make the LSTMs models more competitive in future work**
>
> We thank the reviewer for comments. We address them point-by-point below.
>
> **Main**
> 1. **"Attention can be integrated into recurrent neural networks..."**
> We agree with this point. Our work does indeed suggest that attention is a key resource for performance on our task. We think that using more competitive architectures (e.g. memory-augmented, attention-based LSTMs), as opposed to vanilla LSTMs, is an exciting future work. Given the time constraints for current revision, we have simply highlighted this future line of investigation in the Discussion section of the current version.
>
> 1. **"The copy mechanism is a standard architecture..."**
> We thank the reviewer for the suggestion. Similar to the memory-augmented LSTMs, we agree that this is another candidate architecture worth testing. In our work, we sought to focus on commonly studied causal transformers. As pointed out in another reply, our rationale for starting with causal (i.e. left-to-right) LM architectures was that, in computational psycholinguistics, causal LMs -- as opposed to bidirectional models -- are typically considered of greater interest as language _processing_ models. This is because humans have a very limited preview of the right-side context in language. Given that we drew inspiration from studies of human memory, the choice of causal LMs was better motivated. But the reviewer is completely correct that by testing the insertion of a COPY mechanism in both LSTM and Transformer models, we can test whether this is a key functional property.  For the current revision, we have added a reference to the COPY architecture papers to our Discussion and we now highlight this as a potential line of investigation.
>
> 1. **The robustness of transformers to intervening text...**
> Thank you. This is an intriguing possibility. Note that for revision, we decided to evaluate an additional new vignette in which the original prompt string has the same tokens, but these are randomly permuted (“When read she got again, back the list she:”). We show that the results remain quantitatively unchanged for the shuffled prompt for both the GPT-2 (Appendix, Fig. 9 D) and the wikitext-103 transformer (Appendix, Fig. 10 D). Given that the results generalize even to ill-formed text sequences, the ability to follow copy instruction from _linguistic instructions_ is likely less important presently. However, we now emphasize the role of the architectural factors more in the Dicussion section and propose this as a point in our discussion for the current version as suggested by the reviewer.
>
> **Minor**
> 1. **"I suggest adding "absolute value" for the part involving GPT-2 increased 8%."**
> Thanks, but we refer to approximate values and the exact values for all comparisons are reported in the tables in the appendix.
> 2. **Holtzman et al. (2020) citation**.
> The reviewer is correct. Our citation is indeed ambiguous. We have corrected it such that it is clear that Holtzman et al, 2020, work on repetition mitigation strategies.
> 3. **missing references ("Similar limitations of LSTMs...")**.
> Thanks. We have added the respective citation for the works closer to the statement.

---

> > ### Comment · Reviewer_ax86 · 2021-11-22
> > **Thanks for addressing in detail all my comments**
> >
> > Thank you very much for authors for having responded and addressed all my comments.
> >
> > Now one point just for my curiosity (on the intriguing possibility one): I totally agree that "Given that the results generalize even to ill-formed text sequences, the ability to follow copy instruction from linguistic instructions is likely less important presently". But what is the intuition that transformers learn better to copy? What has the transformers learned in fact? My humble intuition is that transformers indeed learn to "understand" from the text that the task consists of copying previous words, just that its "understanding" might be quite robust to the ill-formedness of the words?

---

> > > ### Author Response · Authors · 2021-11-23
> > > **Transformer has learned to use specific tokens as cues for verbatim retrieval**
> > >
> > > Thank you for this follow-up point.
> > >
> > > Transformer learning or “understanding” the linguistic copy instruction (however robustly) is a plausible hypothesis. Another possibility is that the transformer has learned to use tokens at specific time-steps (e.g. colon, first token in the list...) as _cues_ indicating that verbatim parts of context are likely to repeat at the upcoming time-steps. We can think of these tokens as high "signal-to-noise" tokens (e.g. a colon, "again" etc.) in that they are reliably indicating which verbatim parts of past context need to be retrieved for next time steps.
> > >
> > > As the cue-token representation is fed through the transformer layers -- this pushes the space of attention weights (these transformations/weights/parameters are learned) to selectively weigh and retrieve from memory those tokens in the past context that "have to be copied" (put differently: those tokens that are likely to repeat due to the presence of the cue-token). These attention-weighted context tokens will receive higher output probabilities (and hence, sampling from such a distribution will likely result in repetitive output).
> > >
> > > In some sense, the sketch above can be viewed as similar to reviewer's intuition; in the sense that the "cue" can be thought of as the "copy instruction".

---

> > > > ### Comment · Reviewer_ax86 · 2021-11-23
> > > > **Thank you for the discussion**
> > > >
> > > > I have adjusted my score to 5.
> > > >
> > > > For the paper that reviewers reviewed, I still think that it doesn't match the ICLR paper standard. However, I think the authors have actively engaged in the discussion and have addressed the all comments, at least in the short term; in the long term, I hope the paper can be accepted elsewhere with a better writing (by improving reference and discussion) and maybe some more experimental analysis to highlight authors' main contributions.

---

### Official Review · Reviewer_WHFY · 2021-10-27

**Correctness:** 3
**Technical Novelty And Significance:** 3
**Empirical Novelty And Significance:** 3
**Recommendation:** 6
**Confidence:** 3

**Main Review:**

This paper studies a very important question: to what extent can NLP models retrieve past input? Memorization is an important skill for communicating in natural language, and I am not familiar with too many works that try to measure it explicitly (see [1] for one related example). The setup proposed by the authors generally makes sense, and the results, at least with respect to GPT-2, are non-trivial: GPT-2 memorizes list1 verbatim, even after as many as 435 words. Overall these results will be of interest to the ICLR community.

My main concern about this paper relates to the LSTM baseline and the interpretation of its results. The authors basically present one positive result in this paper (GPT-2, though arguably the small transformer is also somewhat positive), and one negative result (the LSTM). They then use the negative result to say (or at least hint) something general about LSTMs. I am not sure I would jump to conclusions so quickly here. Take fig2 for instance. LSTMs and Transformer trained on Wiki103 exhibit a rather similar trend: we see very small differences between novel, permuted and repeated. The main difference is the loss value, which is higher for LSTMs. But this does not indicate that transformers memorize better than LSTMs, just that the transformer LM is of higher quality (by the way, what was the perplexity of the LSTM model?). Further, the LSTM was trained with 2 layers. The depth analysis of transformers shows that shallow transformers (<= 3 layers) are unable to memorize, so this factor alone could explain why the LSTM model was unable to memorize. On a related note: how many layers did the wiki103 transformer presented in figures 2-4 have? They surely don't have the same number of layers as the LSTM (as the authors didn't train any transformer with 2 layers). There are other confounding factors that make it hard to compare against the two model, e.g., the tokenization (the LSTM uses words, while the transformer uses BPEs). This also relates to a somewhat surprising choice by the authors (as far as I understand): to match the vocabulary sizes, which is hard to interpret given the different tokenization.


To summarize, here are the concrete questions I think should be addressed in the next version:
1. What is the perplexity of the LSTM model?
2. How deep is the transform that was used in the main experiment?

In addition, I would consider reframing the story about LSTMs.

References:
[1] https://arxiv.org/abs/1805.11653


**Summary Of The Paper:**

This paper studies the short-term memory of NLP models. The authors design a memorization task of the format: <prefix> <list1 of words> <infix> <list2 of words>, where list1 and list2 are either identical, a permutation of one another, or unrelated. They train several models (an LSTM and several transformer variants, including GPT-2) with the language modeling objective, and then measure the median LM loss of each model across all the words on list2. They find that their LSTM does not show any memorization skills, the medium size transformer is able to memorize to a certain extent, and that GPT-2 memorizes almost perfectly, at least when list1=list2. Further analysis studies the factors that influence this memorization, including the length of the list, the length of the infix, and the depth of the model.


**Summary Of The Review:**

Pros:
A very important research question, a nice experimental setup, interesting results regarding GPT-2.

Cons:
Potentially inaccurate conclusions regarding the connection between LSTMs and transformers due to various confounding factors.

---

> ### Author Response · Authors · 2021-11-22
> **Surprisal and relative change in surprisal suggest substantial differences in the transformer and the LSTM model, nevertheless, testing better quality LSTM LMs will be important future work**
>
> 1. **"What is the perplexity of the LSTM model?"**
> We agree with the reviewer that there are differences in overall perplexity, and that these differences are important. We now report these in the updated version. The test-set perplexities for the 400-dimensional and the 1600-dimensional LSTM are 108.5 and 79.6, respectively. It is not clear whether it is the LM quality or the LM architecture that leads to performance differences on our task, and we have also added a paragraph to the Discussion to highlight this limitation.
> We are not aware of better-performing LSTM checkpoints that we could use at this point, but are working on addressing this for future experiments. We do not agree that the Transformer and LSTM exhibit the same behavior when trained on Wikitext 103; on the contrary, the data indicate substantial differences between the models. For the very first token of the list, the Transformer perplexity for the Repeated token is about half of the perplexity of a Novel token (Figure 2B); whereas, for the LSTM model, the perplexity of the first token is the same for Repeated and Novel lists (Figure 2A). Thus, the Transformer  is retrieving the first word in the list, while the LSTM is not. For the later words in the list, the Transformer does not perform as well, but overall there is still a ~20% decrease in perplexity for Repeated vs. Novel lists (Figure 3B), whereas the LSTM shows ~0% decrease in perplexity for Repeated vs. Novel lists (Figure 3A).
>
> 1. **How deep is the transform that was used in the main experiment?**
> We used the  12-layer transformer, and the reviewer is correct that the number of layers remains unmatched between the Transformer and LSTMs models. We added a clarification statement in the methods section that the (GPT2-matched) 12-layer model is reported in the main figures. We also now acknowledge this limitation in the discussion section, and  recommend a comparison of depth-matched models for future work.
>
> 1. **Confounding factors -- tokenization. "This also relates to a somewhat surprising choice by the authors...""**
> This is a valid point. We thought that the best way to keep the parameter numbers as close as possible and keep the output loss as comparable as possible was to also keep the vocabulary dimensions matched. Although we do not expect this design parameter to account for the differences we observed, we now note this potential limitation in the Discussion section of the manuscript, and would consider this as a valuable future experiment.

---

> > ### Comment · Reviewer_WHFY · 2021-11-23
> > **Thank you for your response**
> >
> > I think my main comment remains: the high level framing of the paper as comparing LSTMs and Transformers should be reconsidered. The comparison between the two architectures suffer from many confounding factors, so the conclusions that one model is superior to the other is not supported.

---

### Official Review · Reviewer_M2A9 · 2021-11-01

**Correctness:** 3
**Technical Novelty And Significance:** 3
**Empirical Novelty And Significance:** 3
**Recommendation:** 6
**Confidence:** 4

**Main Review:**

Strengths:
- The experimental paradigm is interesting and translates well to this setting
- The experiments are well-executed, and many alternative hypotheses are considered. Their conclusions seem justified from the experiments.
- The conclusions are interesting, particularly those contrasting LSTMs with Transformers.

Weaknesses:
- The conclusions apply within the (interesting) experimental paradigm, but it's not clear how they translate to natural language and other tasks. For example, the experiments convinced me that Transformers are better than LSTMs at recalling verbatim context (which makes sense, given the way self-attention works), while LSTMs only maintain a gist of the context that they've processed. However (thinking about practical applications, which admittedly may be beyond the scope of this work), it's not clear to me when "verbatim recall" is necessary vs. it's better to have a semantic gist of context. Perhaps some more discussion about this would be useful.

**Summary Of The Paper:**

This paper studies how LSTMs and Transformers represent prior context. In particular, this work adapts benchmark tasks for human working memory to neural language models. In this paradigm, the model is presented with <preface text> <a list of words> <intervening text> <the same list of words>, and perplexity / surprisal is measured for each of the words when the list of words appears a second time at the end of the input. Their results lead to a variety of insights about how LSTMs and Transformers trained on varying amounts of data with varying amounts of depth handle context.

**Summary Of The Review:**

This paper thorough studies LSTMs and Transformers on a recall-based probing task. The work raises several interesting conclusions about LSTMs and Transformers, and they are well-supported by the experiments. However, it is somewhat unclear to me how these experiments might translate to other settings, and I'd like to hear more from the authors about this. Overall, I thought the paper was interesting and well-executed, if a bit narrow, and would be an interesting paper to the ICLR audience.

---

> ### Author Response · Authors · 2021-11-22
> **Semantic gist can be an efficient heuristic when memory channel is limited**
>
> 1. **The conclusions apply within the (interesting) experimental paradigm...**
> We appreciate the call for conceptual clarity. We do not see the capacity to perform verbatim retrieval/indexing and the coarser semantic gist as necessarily mutually exclusive processes.
> On the one hand, semantic gists might well be an efficient strategy when the potential memory capacity is limited (i.e. compressed in a single hidden state vector as in an RNN). Humans can use high-level gist-like heuristics to deal with rapid language processing. For example, it is well known that human recall of language material will generally be better when the items to be recalled contain or are embedded within semantic/conceptual structure (e.g. recalling well-formed sentences vs. arbitrary word lists; Potter, 2012, https://doi.org/10.3389/fpsyg.2012.00113). Further, it is known that human participants will frequently maintain a “good-enough”, (gist-like) representation (Ferreira & Patson, 2007, https://doi.org/10.1111/j.1749-818X.2007.00007.x) rather than track the detailed (indexable) representations of past tokens.
> On the other hand, repetition is also part of language, as people do repeat tokens and passages of text from earlier time points. More generally, the process of generating an indexable representation of earlier tokens is necessary for predicting language because such orderings are used by humans. For example, people make use of parallelisms: “I just finished a whirlwind trip to France, Germany and Japan. Traveling from Paris to Berlin to Tokyo, it was Bonjour on Monday, Gutentag on Tuesday and Konnichiwa on Wednesday!” We have now added these additional discussion points to the paper (paragraphs 2 and 3 in the Discussion).

---

### Official Review · Reviewer_r2TC · 2021-11-02

**Correctness:** 3
**Technical Novelty And Significance:** 2
**Empirical Novelty And Significance:** 3
**Recommendation:** 5
**Confidence:** 4

**Main Review:**

Strengths:

 - The question explored by the paper is interesting and important given the ubiquity of these models. The extent to which models such as GPT-2 can retrieve prior information looks very interesting and the difference in performance between Transformers and LSTMs is also worth noting.

 - The results also add to possible reasons why in the space of large-scale LMs, Transformers have been more successful than LSTMs.

 - The paper is very well written and the narrative is clearly presented.

Weaknesses and Questions:

 - All the experiments seem to be intended on LMs trained on large/medium size datasets particularly Wikitext-103. Since the overall goal is to understand the ability of these models to access prior context, it would be interesting to compare models trained on smaller datasets or maybe even synthetic datasets, to check whether LSTMs perform worse/comparable/better than Transformers in that setting. It seems a bit surprising that LSTMs perform so poorly and it could be useful to characterize, at what stage the difference in performance between Transformers and LSTMs occur.

 - It seems like the dataset on which the models are trained also plays a significant role, looking at the difference in results between GPT-2 and other transformer results. Do you think the results for LSTMs could differ if trained on some other larger dataset? I see that when both the models are trained on the same Wikitext dataset, LSTMs perform poorly compared to Transformers but I wanted to understand why GPT-2 performs much better, if it is because of the dataset, and whether that could also affect LSTMs.

- The experimental design seems a bit weakly motivated. It could be useful if the authors could discuss why the setup is more suitable for evaluating the ability of LMs to access memory as compared to some other arbitrary choices.

  - The approach to compare LMs based on reduction in surprisal has been used by several recent works (as cited in the paper) but it is not clear if it is the best way to test a model's ability to access memory. Did you consider any alternatives? Any reason why this seemed like the best option?

  - Also, why did you choose to go with noun lists and also in particular those noun lists? Since some models like GPT-2 did quite well with noun lists of length 10, I was curious to what extent they could maintain such a list. Did you try longer lists?

  - In section 3.4, why did you consider the median surprisal?


The first experiment in section 4.4 does not make sense to me. How or why would a randomly initialized Transformer (or any LM model) reproduce (or assign a high probability to) a list of nouns based on English-like prompts?

It is mentioned that the depth of a Transformer seems to be a key attribute. Does the number of layers also affect LSTMs or does their performance saturate after a few layers?


**Summary Of The Paper:**

The paper attempts to investigate the ability of Transformer and LSTM based LMs to retrieve information about the prior context. To do so, they provide the LMs some prompts, a list of nouns and check their ability to reproduce the list by comparing the probabilities assigned to the same list of nouns (checking the reduction in surprisal).

The premise of the work is that making use of prior context is an important functionality of language models and while several prior works have demonstrated that LMs do so to capture certain types of dependencies, their work aims to understand the extent to which Transformers and LSTM based LMs can retrieve words from the prior context.

Via their experiments, they try to evaluate three things: (1) How well can such LMs reproduce the noun list from memory verbatim, (2) the effect of intervening tokens on an LM's ability to access, and (3) whether the ability to access memory depends on semantic coherence of the prior information.

They considered two versions of LSTM models and a few versions of Transformer models including GPT-2. Based on their results, it seems like Transformer based LMs performed consistently better than LSTMs on the defined task. The GPT-2 model performs much better than others. They also did some experiments to evaluate the role of attention weights by comparing with a model with shuffled attention weights.

**Summary Of The Review:**

Overall it is an interesting and well-written paper trying to explore an important question with some sound experiments. At the same time, it seems to have certain weaknesses in the experimental setup and design which diminishes the conclusiveness of some of their results.

---

> ### Author Response · Authors · 2021-11-22
> **Dataset size plays an important role, however, architectural elements (e.g. attention) may likely be needed in LSTMs to solve task (1/2)**
>
> 1. **"All the experiments seem to be intended on LMs trained on large/medium size datasets...""**
> We appreciate this proposal. Our results do strongly suggest that the role of dataset size is crucial. Our hesitance with respect to training on smaller/synthetic datasets is twofold. First, our results suggest that a smaller dataset leads to worse performance (smaller effects) in both models. It is therefore reasonable to think that experiments on much smaller datasets would likely further dilute the observed retrieval effects. A potentially powerful idea would be to adapt the existing models on additional synthetic datasets (e.g. that include lists of nouns) and see whether retrieval capacity can be encouraged despite the overall smaller dataset size. As we point out in the Discussion section, LSTMs have been shown to struggle on copy tasks involving longer sequences and thus augmented architectures might be needed for better performance. We have highlighted this as a future avenue in the Discussion section.
>
> 1. **"It seems like the dataset on which...""**
> Thanks for pointing this out. Based on the results so far, our current hypothesis would be that — in absence of additional architectural features, larger LSTMs would remain limited in retrieving specific word ordering information. This is because even when the LSTM was trained on a two times larger dataset compared to the first checkpoint (80 million tokens, Fig. 6, Appendix), there was still not evidence of it exhibiting retrieval of word ordering as the wikitext-103 transformer did. We used the Wikitext-103 dataset as this is a commonly used dataset and many existing models are benchmarked against it. Second, it is likely that the dataset size alone does not determine the performance on this task --- when we permuted the attention weights of the Wikitext-103 transformer, it no longer exhibited the capacity to distinguish and retrieve ordering of the nouns in the lists. To us, this suggests that attention, in addition to dataset size, might be a crucial component. We now explicitly highlight this -- the future potential in studying attention- or memory-augmented LSTMs in the Conclusions section.
>
> 1. **"The approach to compare LMs...""**
> Given the prior work and given our interest in seeing whether or not the language modeling objective could push the systems towards developing memory capacity (verbatim recall in our case), we decided to quantify memory retrieval as a relative reduction in loss (i.e. surprisal) -- as this is the quantity directly minimized during training. As a more explicit measure of verbatim recall we initially considered looking at the text generated after the retrieval prompts (e.g. how often does the model generate the exact sequence to be retrieved?). However, as will be familiar, text-generation introduces a number of additional choices (sampling strategy, logit distribution scaling etc.). As we were unable to resolve the uncertainties arising from this generative evaluation scheme, we decided to pursue the more constrained surprisal metric.
>
> 1. **Why did you choose to go with noun lists...**
> We chose to start with noun lists because nouns often individuate entities in the world, and thus are likely to be a necessary component in almost any kind of representation of prior context. Moreover, lists of  nouns have provided the target lists in many studies of human memory and are publicly shared (http://memory.psych.upenn.edu/files/wordpools/nouns.txt). We did not consider longer lists at this point; with longer lists (> 15 tokens) the number of sequences we can construct becomes limited due to limited number of tokens in the  categorized lists of nouns (n. of nouns per category which at the same time are part of the LSTM vocabulary is limited). Nonetheless, because the GPT-2 model of Radford et al showed almost no decrease in performance with increasing list length, we would hypothesize that this model could recall lists up to the length of  its entire input buffer, and we will test this in future work. We also agree with the reviewer that testing other types of words (e.g. verbs, adjectives, and function words) is a natural next step.
>
> 1. **The experimental design seems a bit weakly motivated...**
> Indeed, a randomly initialized Transformer is normally not expected to perform the task. However, reports in the past have shown that the architecture alone (i.e. randomly initialized networks) can still be surprisingly useful in extracting features from the input data (see Saxe et al., 2011, ICML 2011 for example with convolutional networks). We therefore sought to ascertain that the performance on our task does indeed require learning (language modeling objective) and that the architectural features per se are not sufficient.

---

> > ### Author Response · Authors · 2021-11-22
> > **Dataset size plays an important role, however, architectural elements (e.g. attention) may likely be needed in LSTMs to solve task (2/2)**
> >
> > 6. **"It is mentioned that the depth..."**
> > Correct, the transformer performance did improve with more layers. We do not have a pre-trained LSTM with > 2 layers currently available. We are not aware of available LSTM checkpoints. Potential solution to this is to train a custom LSTM model with more layers, however, we were unable to perform this in the timeframe of this revision. We now highlight this limitation in the discussion section (paragraph 4).

---

> > > ### Comment · Reviewer_r2TC · 2021-11-30
> > > **Thank you for the detailed response**
> > >
> > > I have read the authors' response and while some of my minor concerns have been convincingly addressed, I would still like to keep my score unchanged.

---

### Official Review · Reviewer_Tpb6 · 2021-11-03

**Correctness:** 2
**Technical Novelty And Significance:** 2
**Empirical Novelty And Significance:** 2
**Recommendation:** 3
**Confidence:** 4

**Main Review:**

**Strengths**

- The paper investigates an interesting task.
- It is understandable, well structured and well written.
- Experiments conducted with varying list sizes, different list types (arbitrary vs. coherent words), and different sizes and content of intervening text to provide insights.
- The work is reproducible as the code is provided.

**Weaknesses**

- The data is synthetic and developed based on predefined templates that represent an extremely narrow sequences of words in the following format:   *[she created the following list of words:] ["1st list of nouns separated by commas."] [she took a break blah blah.] [when she got back, she read the list again:] ["2nd list of nouns"].*    In addition, this way of text generation can potentially lead to implausible texts. For example, in several samples provided in the appendix, it's not clear whether "got back" in the above template actually refers to the first list or to the things that happen in the intervening text. Given these reasons, I think the reported results are less likely to be generalizable to real world datasets.

- Although language modeling has been used as a motivation for investigating memory in neural models, the task formulation is very different from language modeling, which is the task of predicting the probability of a sequence of words occurring in a language, not necessary predicting a sequence of nouns separated by commas. LMs should be able to learn syntactic structure such as subject verb agreement, long-range dependencies among words, the role of function and content words, etc.

- It's not clear if the task is truly evaluating models' ability in retrieving/recalling words from context. The model may accurately predict some of the words in the 2nd list due to their correlation with the intervening words but not because of their occurrence in the first list.

- A limited number of words (~ 230 unique words) and lists (~ 230 lists of length <=10 words) are used in experiments. These number of words and sequences are of small scale and may be simply memorized by the model.

- Surprisal of each list is estimated by the "median" instead of "average" of the surprisal scores obtained for non-initial nouns in the list. Although median might be a better metric than average in many settings, here, memory should be evaluated against the entire list but not one of the words in the lists.

- The number of parameters significantly differs between transformers and LSTM networks used in experiments, which naturally affects their performance and makes the results not directly comparable.

- It's not clear why transformer models are compared against vanilla LSTMs but not bidirectional LSTMs with attention.







**Summary Of The Paper:**

This paper argues that language models (LMs) require mechanisms to retain information about context words for better next word prediction. Several free recall experiments have been conducted to determine if such models can exactly retrieve/recall words from context. Specifically, different LMs are given sequences that contain texts in the following order: a preface text, a list of $n$ target nouns ($n\leq 10$), an intervening text, a short prompt string, and a second list of nouns, which are either identical to the first list or different from it (i.e. similar words but permuted or completely different words). Reduction in surprisal from the first lists to the second lists is used as a metric to quantify and compare memory of different LMs. Surprisal of each list is estimated by the "median" of the surprisal scores obtained for non-initial nouns in the list. Several transformer models and LSTM-based language models of different sizes are compared in terms of reduction in surprisal. Results show that transformer models are superior in retrieving word identity and order from small/long-range intervening text (from a few to more than 400 tokens). Also more training data and greater model depth lead to better performance in case of transformer models.

**Summary Of The Review:**

This submission investigates the interesting question of memory in language models, but the task formulation, data and experimental setup can be improved and therefore I don't recommend the paper for publication at this time.

---

> ### Author Response · Authors · 2021-11-22
> **The ability of LM's to predict repetitions manifests the ability to index ordered sets showing how LMs satisfy the objective (1/2)**
>
> 1. **"The data is synthetic and developed..."**
> We agree that the generalization concern with respect to our prompts is a concern. To address this, we decided to evaluate an additional new vignette in which the original prompt string has the same tokens, but these are randomly permuted (“When read she got again, back the list she:”). We show that the results remain quantitatively unchanged for the shuffled prompt for both the GPT-2 (Appendix, Fig. 9 D) and the wikitext-103 transformer (Appendix, Fig. 10 D). Given that the results generalize even to ill-formed text sequences, we expect them also to generalize to other well-formed prompts.
> 1. **"Although language modeling has been..."**
> We agree with the reviewer’s point that short-term memory (operationalized here as verbatim retrieval) is only one process contributing to  language comprehension and prediction. We, however, disagree that our paradigm is adversarial to the language modeling objective. Language models are trained with the objective of predicting the next word, but this does not specify _how_ they achieve this objective. Indeed, a long-standing debate in the cognitive sciences concerns the question of whether and how humans use domain-general   working memory systems in the service of language processing (Baddeley, 2003, 10.1016/S0021-9924(03)00019-4). Here, we try to understand which kinds of functional architectures the model may acquire in the service of satisfying the LM objective. For example, prior work has shown that LMs learn some aspects of syntax, but there is nothing about the LM objective that requires this functional property to emerge. In this work, then, the ability of models to predict repetitions of previous sequences is not of interest in itself, but rather as an _index of the memory processes_ that the model may have acquired, reflecting their ability to specifically index an ordered set of prior tokens.
> 1. **"It's not clear if the task is truly..."**
> This is a valid concern in general, but our paper included manipulations that ruled out this interpretation. If the correlation with intervening words (e.g. word co-occurrences) was driving the performance of LMs as the reviewer suggests, then memory retrieval performance should also be observed in the control condition (“novel lists”) for which we used the same nouns as in other conditions, but these nouns did  not appear earlier in the context. We did not observe memory retrieval in this condition:  novel tokens -- pink in our plots (e.g. Fig. 3) -- receive 0% surprisal reduction on average. Thus, the surprisal effects that we reported are indicative of retrieval of tokens from their prior occurrence, and not from their co-occurrence in the target list.
> 1. **"A limited number of words..."**
> We agree with the reviewer that, in future work, these results should be extended to other kinds of tokens. The paradigm is flexible and such extensions are straightforward to perform if additional pools of nouns (or other parts of speech) are provided in the future. However, we do not understand the concern about memorization, as all of our word lists are “out of sample”, i.e. the models have not seen the word lists before and they are not trained while performing the task. Moreover, as noted in our response to the reviewers’ previous concern, the models cannot retrieve the lists at all unless it occurs earlier in the context: thus, the models are retrieving information from their context representations on each run, rather than retrieving them from memories encoded in fixed pre-trained weights.
> 1. **"Surprisal of each list..."**
> We should have clarified: our initial report was inaccurate we averaged across token position and then used the median to aggregate across N=320 sequences. We opted for median in this case because it is more robust to outliers (e.g. potential trials with high/low average surprisal) that could drive our estimation of central tendency and by consequence our experimental effect. To address this concern explicitly, we now show in the Appendix (Fig. 10), hold even if we chose the mean as the estimator across sequences. It is not entirely clear to us at present what is suggested by “memory should be evaluated against the entire list but not one of the words in the lists” as this is exactly what we achieve by aggregating over the entire list.
> 1. **"The number of parameters..."**
> However, note that we also tested a larger LSTM (Appendix, Figure 6, right) which had 132 million learnable parameters. This is slightly more than the 12-layer GPT-2 small (124 million learnable parameters). Despite the larger parameter count, however, the 132-million-parameter LSTM did not perform retrieval to the same extent as the 124-million-parameter transformer did.

---

> > ### Author Response · Authors · 2021-11-22
> > **The ability of LM's to predict repetitions manifests the ability to index ordered sets showing how LMs satisfy the objective (2/2)**
> >
> > 7. **"It is not clear why bidirectional LSTM..."**
> > We thank the reviewer for the suggestion. In the timeframe of the current revision it was not possible for us to train additional models, but we would welcome any pointers to pre-trained LSTM models with attention (ideally available in PyTorch). Our rationale for starting with causal (i.e. left-to-right) LM architectures was that, in computational psycholinguistics, causal LMs are typically considered of greater interest as _processing models_ compared to bidirectional models. This is because humans have a very limited preview of the right-side context in language. Given that we drew inspiration from studies of human memory, the choice of causal LMs, as opposed to bidirectional LMs, was better motivated. The paradigm itself, however, certainly doesn’t not preclude testing bidirectional models and would be considered interesting future work.

---

> > > ### Comment · Reviewer_Tpb6 · 2021-11-29
> > > **Some points are cleared up!**
> > >
> > > Although some of the questions are addressed by authors, I keep my evaluation unchanged. This is a good first step toward understanding retrieval/recall of context words by LMs, but the contribution would be more solid with better task formulation, generalizability analysis, and experiments with real world datasets.
> > >
> > > **1. Use of synthetic data.**
> > >
> > > I think synthetic text generation can simply result in implausible texts and I suggest authors to use real world datasets to address the generalizability concern. In addition, I don't see how snuffling prompts of any kind can illustrate model generalizability.
> > >
> > > **2. Language modeling paradigm.**
> > >
> > > I agree with authors that the "training" paradigm is not adversarial to the language modeling objective. However, testing through synthetic data -- especially in case of random sequences of nouns -- is adversarial to the language modeling objective. Maybe, these models *should* prioritize learning syntactic structures or long-range semantic dependencies over memorizing words.
> > >
> > > **3. Task formulation and potential effect of word co-occurrence.**
> > >
> > > In terms of task formulation, there can be lists of novel words that lead to considerably higher reduction in surprisal than the first/original list; such lists can be created based on n-gram models (co-occurrence) or model output (e.g., words with very low surprisal that do not occur in the first list). Such sequences can be created and it's not clear how they can be explained in the context of the present work.
> > >
> > > **4. Limited number of words.**
> > >
> > > By memorization, I meant word sequences in real-world datasets that are not out-of-sample and can be memorized by models.
> > >
> > > **5. Computation of surprisal.**
> > >
> > > Thank you for clarifying.
> > >
> > > **6. Number of model parameters.**
> > >
> > > Thank you for clarifying.
> > >
> > > **7. comparison against Bi-LSTM.**
> > >
> > > Some good references (not necessarily in pytorch):
> > > - https://aclanthology.org/E17-1096.pdf
> > > - https://openreview.net/pdf?id=BydLzGb0Z
> > > - https://openreview.net/forum?id=HJ0UKP9ge
> > > - https://arxiv.org/pdf/1602.06064.pdf
> > > - Original paper on bi-lstms: Mike Schuster and Kuldip K. Paliwal, Bidirectional Recurrent Neural Networks, 1997.
> > > https://maxwell.ict.griffith.edu.au/spl/publications/papers/ieeesp97_schuster.pdf

---

### Decision · Program_Chairs · 2022-01-20

**Decision:**

Reject

**Comment:**

This paper explores the memorization of tokens in prior context in LSTM and Transformer based language models. While all reviewers agree this is an interesting and important direction worth studying, they raise several concerns about the validity of the experimental setup and the conclusions drawn about LSTMs. Primarily the shallow depth of the LSTM architecture seems to confound the main conclusion about their inferiority (Reviewer WHFY). Further exploration about what makes transformers better (e.g. attention) is also important to provide a more complete picture (Reviewer ax86). Other concerns include the use of synthetic data (Reviewer tpb6), a limited number of noun lists (Reviewer r2TC) and the lack of discussion about the practical significance of verbatim recall (Reviewer M2A9). Overall, while the paper takes a step towards an important insight about pretained LMs, it needs to be polished further and hopefully can be published at a future conference.